# Dietary ω−3 polyunsaturated fatty acids (PUFAs) reduce cholesterol-driven non-small cell lung cancer (NSCLC) progression in mouse models of disease

Philipp Harre[1], Katja Hohenberger[1], Susanne Krammer[1], Zuqin Yang[1], Patrick Tausche[1], Jonas Willar[1], Denis I. Trufa[2,3,4], Mircea T. Chiriac[5], Carol I. Geppert[3,4,6], Manfred Rauh [7], Kai Hildner[3,4,5], Arndt Hartmann[3,4,5], Horia Sirbu[2,3,4] & Susetta Finotto [1,3,4,8] ✉

## Abstract

**Background** Non-small cell lung cancer (NSCLC) is the leading cause of cancer mortality, and most patients fail to respond to immune checkpoint inhibitors (ICIs). Nutritional factors, particularly dietary lipids, influence cancer progression and immunity. Cholesterol can promote tumor growth and immune evasion, whereas ω−3 polyunsaturated fatty acids (PUFAs) may exert anti-inflammatory and immunostimulatory effects. We aimed to determine how dietary lipids influence NSCLC progression and whether ω−3 PUFA supplementation can enhance the efficacy of PD-1 blockade.

**Methods** Serum lipid profiles from 152 NSCLC patients were analyzed for associations with prognosis. Female murine NSCLC models were fed diets enriched in cholesterol, saturated fats, or ω−3 PUFAs, with or without anti-PD-1 therapy. Tumor growth, immune infiltration, cytokine production, and epithelial–mesenchymal transition (EMT) markers were assessed by flow cytometry, ELISA, and gene expression analysis.

**Results** In patients, elevated serum triglycerides correlate with poor outcomes, and tumor cholesterol metabolism links to EMT marker expression. In female mice, cholesterol-rich diets accelerate tumor growth, increase EMT-associated genes (SNAIL, Vimentin), and elevate pro-tumoral cytokines (IL-6, IL-10, IL-17A). ω−3 PUFA diets reduce tumor burden, lower immunosuppressive cytokines, decrease regulatory T cells, and enhance cytotoxic T cell activity. Combining ω−3 PUFAs with anti-PD-1 therapy synergistically suppresses tumor growth and improves antitumor immune responses.

**Conclusions** Dietary lipids modulate NSCLC progression via metabolic, inflammatory, and immune pathways. ω−3 PUFA supplementation counteracts cholesterol-driven tumor promotion and augments PD-1 blockade efficacy, supporting dietary modulation as a complementary strategy to improve immunotherapy outcomes in NSCLC.

## Plain language summary

We study how dietary fats affect lung cancer and its treatment. Lung cancer is the leading cause of cancer deaths, and many patients do not respond well to modern immunotherapies, a type of cancer treatment. Cholesterol can make tumors grow faster and weaken the immune system, while ω−3 fatty acids (found in fish and some plants) are thought to have protective effects. We analyzed blood and tumor samples from lung cancer patients and tested different diets in mice, with or without immunotherapy. We find that cholesterol fuels cancer growth, while ω−3s slow it down, strengthen immune defenses, and make immunotherapy work better. These results suggest that eating more ω−3 fatty acids could help support cancer treatment, though clinical trials are needed to confirm this finding.

Lung cancer is the second most common cancer type worldwide, with 12% of all new cancer diagnoses and has the highest lethality as it is accountable for about 21% of all cancer-related deaths in the United States[1]. As therapeutic strategies for lung cancer have improved throughout the years regarding surgery, radiotherapy, chemotherapy, and immunotherapy, the 5-year survival rate still remains poor with a median of 14%[1]. Substantial

side effects and restricted response rates of current anti-tumor therapies and the development of escape mutants represent the main challenge.

In addition, lung cancer is increasing worldwide, one of the reasons being diet composition. We recently analyzed a large cohort of patients with adenocarcinoma and found that patients with high blood triglyceride levels have a worse prognosis compared to patients with high total cholesterol

levels. Moreover, we found a direct correlation between serum triglyceride levels in patients with lung adenocarcinoma and their BMI. Total serum cholesterol, on the other hand, correlated indirectly with the tumor diameter, indicating that the tumor diameter decreased with increasing total cholesterol serum levels[2].

This is most likely due to the fact that tumor cells accumulate intracellular cholesterol and create cholesterol-rich tumor microenvironments (TME), thus stimulating immune-suppressive tumor infiltrating immune cells that drive cancer progression[2–5]. In comparison to healthy cells, previous studies found Sterol Regulatory Element Binding Protein (SREBP) as well as the downstream low density lipoprotein receptor (LDL-R) upregulated in prostate cancer cells[6]. In addition, the key enzyme in cholesterol neo synthesis, HMG-CoA reductase (HMGCR), as well as acetyl-CoA C-acetyltransferase 1 (ACAT-1), have been found upregulated in lung tumor tissue with ATP-binding cassette transporter A1 (ABCA-1) downregulated, thus revealing altered cholesterol metabolism of tumor cells in terms of increased cholesterol uptake and neosynthesis, coupled with impaired cholesterol tumor cellular efflux[2].

Increasing intracellular cholesterol enables tumor cells to evade the immune system by various mechanisms: cholesterol-rich tumor cell membranes impair $CD8^+$ T-cell mediated cytotoxicity and initiate recruitment of circulating tumor cells at metastatic sites[7,8]. At the same time, tumor infiltrating $CD8^+$ cytotoxic T lymphocytes (CTL) become exhausted more rapidly, as the cholesterol rich TME increases endoplasmatic reticulum stress, hence favoring expression of co-inhibitory receptors Programmed cell Death protein 1 (PD-1), Cytotoxic T-Lymphocyte Antigen 4 (CTLA-4), T-cell Immunoglobulin and Mucin-domain containing-3 (TIM-3), and Lymphocyte Activation Gene 3 (LAG-3) on the cell surface of $CD8^+$ CTLs[5]. The neosynthesis of cholesterol via 3-Hydroxy-3-Methylglutaryl-CoA Reductase (HMGCR) is inhibited by statins, drugs commonly used to lower blood cholesterol levels in clinical practice. Simultaneously, administration of statins has shown downregulation of co-inhibitory receptors PD-1, TIM-3, LAG-3, CTLA-4, CD160, T cell Immunoreceptor with Ig and ITIM domains (TIGIT), and CD244 on T-cells, indicating a close connection of cholesterol and immune checkpoints[9]. Restricting the availability of cholesterol via HMGCR inhibiting drugs is associated with reduced cancer-related mortality, amplifying the importance of cholesterol metabolism in relation to cancer[10]. Cholesterol and its metabolites can enhance the ability of cancer cells to invade nearby tissue and metastasize by activating the Nuclear Factor kappa-light-chain-enhancer of activated B cells/ Protein Kinase B/ Peptidylprolyl Isomerase B axis (NFκB/AKT/PPIB axis), as well as increasing Interleukin-6 (IL-6) and Fibroblast Growth Factor 2 (FGF-2) secretion, inducing the expression of Snail family transcriptional repressor 1 (SNAIL) and Vimentin, proteins strongly connected to cancer progression and metastasis[11].

While cholesterol-rich diets are known to cause numerous nutrition-related health issues, polyunsaturated fatty acids (PUFA) — especially ω−3 PUFAs have been shown to have a positive effect on patients' immune status[12]. ω−3 PUFAs constitute a subclass of essential long-chain fatty acids distinguished by the presence of multiple cis double bonds, the first of which is positioned at the third carbon atom from the terminal methyl (ω) end of the hydrocarbon chain. The most biologically significant ω−3 PUFAs include α-linolenic acid (ALA; C18:3), primarily derived from plant sources, as well as eicosapentaenoic acid (EPA; C20:5) and docosahexaenoic acid (DHA; C22:6), both of which are predominantly obtained from marine organisms. Beyond serving as structural components of phospholipid membranes, ω−3 PUFAs function as biosynthetic precursors for specialized pro-resolving lipid mediators and exert direct regulatory effects on inflammatory signaling pathways and oxidative stress responses[13]. Furthermore, direct anti-cancer effects of ω−3 PUFAs were demonstrated in murine models of lung, colon, prostate, and mammary cancer, where tumor growth was slowed down by ω−3 PUFA administration[14–16]. At the same time, ω−3 PUFAs are able to counteract some of the pro-tumoral effects of a cholesterol rich diet by inactivating Phosphoinositide 3-Kinase (PI3K)/AKT pathway through ROS, hindering the proliferation and progression of

cancer cells[17]. There is evidence that the success of classic anti-tumoral therapy such as surgery, radiation, and chemotherapy is improved by dietary supplementation of ω−3 PUFA in terms of overall survival rates and quality of life of patients[18].

Taken together, a close connection of diet and immune status indicates that malnutrition increases inflammatory factors and decreases T-cell-mediated anti-tumor defence. Thus, it is important to identify the patient's nutrition-related issues. The objective of this study was to analyze the influence of a healthier diet on the development of lung cancer by reduction of cholesterol and increased intake of ω−3 PUFA. Furthermore, we aimed to characterize immune system alterations associated with non-small cell lung cancer (NSCLC) in both a patient cohort and a murine model. To this end, we employed quantitative real-time PCR, flow cytometric analysis, and in vivo imaging to examine the samples we acquired.

The lung tumor microenvironment (TME)—a highly dynamic and heterogeneous milieu that surrounds malignant cells in primary or metastatic lung tumors—comprises a complex and reciprocal network of cellular and non-cellular components that collectively influence tumor initiation, progression, immune evasion, and therapeutic response.

In this study, we show that elevated triglycerides in patients associate with poor prognosis and that tumor cholesterol metabolism links to epithelial–mesenchymal transition. In mouse models, cholesterol-rich diets accelerate tumor growth and immunosuppression, while ω−3 polyunsaturated fatty acids reduce tumor burden, enhance cytotoxic immune activity, and augment the efficacy of PD-1 blockade. These findings demonstrate that dietary lipids shape NSCLC progression and that ω−3 PUFAs offer a complementary strategy to improve immunotherapy outcomes.

## Material and methods
### Human subjects and study population
This study was performed at the Friedrich-Alexander-University of Erlangen in Germany and was approved by the ethics review board of the University of Erlangen (Re-No: 56_12B; DRKS-ID: DRKS00005376). A total of 152 patients with NSCLC underwent surgery and provided written informed consent to participate in the study. Patients' confidentiality was maintained.

Tissue samples were taken by the thoracic surgeon from the tumoral area (TU: solid tumor tissue) as well as from the tumor free control area (CTR: at least 5 cm away from the solid tumor) of the surgically removed lung material. Tissue samples were immediately transported to our laboratory and preserved in RNAprotect solution (QIAGEN, Maryland, USA, Cat# 76106) at −80 °C until thawing for RNA isolation (refer to section: Lung cell isolation). Relevant clinical data of the patients, as presented in Table 1, were obtained during hospitalization in the perioperative period through standardized questionnaires and clinical anamnesis. The samples analyzed in this study were collected between July 2012 and December 2020.

Lung cancer diagnosis was confirmed by the Institute of Pathology at the University Hospital Erlangen. Tissue samples were classified for histological subtype, according to the 2015 World Health Organization Classification of Lung Tumors. TNM Staging is based on the proposals for Revision of the TNM Stage Groupings in the Forthcoming (Eighth) (TNM8) Edition of the TNM Classification for Lung Cancer by the International Association for the Study of Lung Cancer (IASLC), issued in 2016. All relevant human data for this study were provided by the Department of Thoracic Surgery and the Institute of Pathology, University Hospital in Erlangen. The individual patients' clinical data are shown in Supplementary Tables S1–4 and are summarized in Tables 1–3.

### Mouse lung cancer model
All mice analyzed in this study were bred and maintained under specific-pathogen-free (SPF) conditions at our animal facility (University Hospital Erlangen, Hartmannstraße 14, Erlangen). All experiments were performed in accordance with the German and European laws for animal

**Table 1 | Clinical data of the patients analyzed in this study**

| Characteristics | Mean | | SEM | SD | N |
|---|---|---|---|---|---|
| Gender | 51.5%F / 48.5%M | | | | 33 |
| Age | 65.39 | ± | 1.628 | 9.355 | 33 |
| Body weight [kg] | 76.94 | ± | 3.341 | 19.194 | 33 |
| Body height [m] | 1.69 | ± | 0.018 | 0.104 | 33 |
| BMI = Weight/Height² | 26.71 | ± | 0.956 | 5.495 | 33 |
| Waist circumference [cm] | 103.40 | ± | 4.118 | 15.949 | 15 |
| Tumor Diameter [cm] | 2.91 | ± | 0.29 | 1.739 | 36 |
| Smoking [pack years] | 37.52 | ± | 5.297 | 28.524 | 29 |
| Albumin presurgical [g/l] | 41.79 | ± | 0.718 | 3.798 | 28 |
| CRP presurgical [mg/l] | 13.43 | ± | 5.63 | 27.594 | 24 |
| CRP postsurgical [mg/dl] | 107.70 | ± | 14.28 | 80.806 | 32 |
| Lymphocytes absolute presurgical [x10³/µl] | 1.86 | ± | 0.119 | 0.671 | 32 |
| Lymphocytes before surgery [%] | 24.61 | ± | 1.628 | 9.208 | 32 |
| FEV1 [%] | 84.30 | ± | 3.505 | 19.195 | 30 |
| FVC [%] | 94.36 | ± | 2.881 | 14.970 | 27 |
| Glucose [mg/dl] | 106.18 | ± | 5.903 | 31.235 | 28 |
| Total cholesterol presurgical [mg/dl] | 179.19 | ± | 7.656 | 39.783 | 27 |
| HDL cholesterol presurgical [mg/dl] | 50.00 | ± | 3.695 | 19.199 | 27 |
| LDL cholesterol presurgical [mg/dl] | 105.59 | ± | 5.351 | 27.805 | 27 |
| Triglycerides presurgical [mg/dl] | 120.96 | ± | 7.802 | 40.540 | 27 |
| Lipoprotein A presurgical [mg/dl] | 17.20 | ± | 3.786 | 19.672 | 27 |

protection and were approved by the local ethics committees of the Regierung Unterfranken (Az 55.2-2532-2-1286-20). Two murine strains were used in this study: BALB/c and C57BL/6J. Six weeks old female mice (C57BL/6J; RRID: MGI:3028467) were purchased from Janvier Labs and they were fed with their corresponding diet 5 days after arrival at our animal facility. Three days before tumor cell injection, the food was changed for the intervention diet group in accordance to the experimental protocol described in Fig. 2A. The mice were fed either with modified arteriosclerosis diet (Altromin, C-1061-1) containing 0.15% cholesterol or no cholesterol rapeseed oil enriched diet (Altromin, C-1060) or both, starting with modified arteriosclerosis diet and then changing the food to low cholesterol rapeseed oil enriched diet. As a control we fed a group of mice with standard food (Altromin, 1324). Relevant information for the diets used are displayed in Fig. 2A and Supplementary Tables S5, 6 respectively. In total, the mice were fed with their respective diets for 33 days. For the induction of non-small cell lung cancer, eight-week-old mice were intravenously injected in the tail vein with 5x10⁵ luciferase-expressing LL/2-luc-M38 cells (RRID: CVCL_5J40) suspended in 200 µl DMEM Medium without supplements (Gibco™ DMEM, high glucose, Thermo Fisher Scientific, cat# 11965092). Tumor load in vivo was quantified by bioluminescence imaging on day 16 post injection. To measure tumor burden, luciferin (0.15 mg/g body weight; Promega, Cat#P1043) was administered intraperitoneally, and the mice were returned to their cages for 20 min. After 15 min, mice were anesthetized with Isoflurane for 5 min and then imaged using the in vivo imaging system (IVIS, RRID:SCR_018621). Luminescence intensity served as a measure of tumor load and is reported as photon per second [p/s].

Experimental procedures involving the BALB/c model were conducted separately and are described in the following section. Six weeks old female mice (BALB/c; RRID: MGI:2161072) were purchased from Janvier Labs and they were fed with their corresponding diet 5 days after arrival at our animal facility. The mice were fed either rapeseed oil enriched diet (Altromin, C-1060) or standard food (Altromin, 1324). Relevant information for the diets

used are displayed in Fig. 2A and Supplementary Tables S5, 6 respectively. In total, the mice were fed with their respective diets for 30 days. For the induction of non-small cell lung cancer eight weeks old mice were injected intravenously with $5 \times 10^5$ of L1C2 cells (RRID: CVCL_4358) in 200 µl DMEM Medium without supplements (Gibco™ DMEM, high glucose, Thermo Fisher Scientific, cat# 11965092) in the tail vein.

Each dietary group was further subdivided into two arms based on anti-PD-1 treatment, resulting in four experimental conditions: standard diet with or without therapy, and rapeseed oil enriched diet with or without therapy, as illustrated in Fig. 6C. Anti-PD-1 antibody (IchorBio, cat#ICH1132, RRID: AB_2921498) was administered intraperitoneally in three doses (150 µg in 200 µl PBS), given on days 13, 15, and 18 following tumor cell injection.

All experiments were performed in female mice to minimize variability due to hormonal cycling in males and to ensure consistency with prior studies using the LL/2 and L1C2 lung cancer models. Sample sizes were chosen based on previous experiments while adhering to ethical principles to minimize animal use. Mice were randomly assigned to experimental groups (diet and treatment conditions) upon arrival at the animal facility, and all interventions (tumor cell injections, diet administration, antibody treatments) were performed according to predefined protocols. Outcome assessments were performed according to the study design, with histological tumor load quantification evaluated in a blinded manner by an independent pathologist. No animals were excluded from the analysis except for those experiencing unrelated health issues (e.g., technical injection failure), which were documented. To reduce potential confounders, animals were age-matched, housed under specific-pathogen-free conditions, and provided with standardized diets and environmental enrichment throughout the study.

**Lung cell isolation**
Lung single-cell suspensions were generated by mincing lung tissue into small fragments, followed by digestion with collagenase (2700 U/ml; Sigma-Aldrich: Collagenase from Clostridium histolyticum; Cat# C98991-500MG) and 150 µl DNase (10 mg/ml; Roche Diagnostics GmbH: DNase I; Cat#10104159001) diluted in 10 ml RPMI medium (Gibco™ RPMI 1640 Medium, Thermo Fisher Scientific, Cat# 21875091). Samples were incubated in a shaker (Edmund Bühler GmbH, Inkubationshaube TH 15) for 45 min at 37 °C. The digested material was passed through a 40 µm cell strainer (Greiner Bio-One, Cat#542040). Cells were centrifuged (10 min, 1500 rpm, 4 °C), the supernatant discarded, and the pellet resuspended in ACK-lysis buffer (0.15 M NH4Cl, Carl Roth GmbH + Co. KG, Cat# P726.2; 0.01 M KHCO3, Carl Roth GmbH + Co. KG, Cat#P748.1; 100 M Na2EDTA, GERBU Biotechnik GmbH, Cat# 1034.1000; dissolved and sterile filtered in deionized H2O; pH=7.2–7.4). After 2 min of incubation, cells were centrifuged again under the same conditions. The supernatant was removed, and cells were resuspended in PBS (anprotec, Cat# AC-BS-0002). To remove fat, the suspension was transferred into a 10 ml pipette and gently dispensed into a Falcon tube, followed by centrifugation (5 min, 1500 rpm, 4 °C). The pellet was resuspended in 10 ml PBS, and viable cell numbers were determined using Trypan blue staining in a Neubauer counting chamber.

For cell culture, $1 \times 10^6$ cells from the total lung suspension were seeded in 48-well plates with Gibco™ RPMI 1640 Medium, Thermo Fisher Scientific (cat# 21875091) supplemented with 10% heat-inactivated fetal bovine serum (FCS) (Sigma-Aldrich, Cat# S0615), 1% Penicillin + Streptomycin (Pen/Strep) (anprotec, Cat# AC-AB-0024), and 1% L-Glutamine (2 mM L-Glutamine, anprotec, Cat# AC-AS-0001). Cells were either left unstimulated or stimulated with 5000 ng/ml of plate-bound anti-CD3 (1 mg/ml, BD Bioscience, Cat#555329, RRID: AB_395736) and 2000 ng/ml anti-CD28 (1 mg/ml, BD Bioscience, Cat#555725, RRID: AB_396068). Cultures were maintained for 24 h at 37 °C and 5% CO2. Supernatants were collected for LEGENDplex™ or cytotoxic assays, while cells were harvested using Qiazol Lysis® Reagent (QIAGEN, Maryland, USA, Cat# 79306) according to the manufacturer's instructions.

All media used for human or murine cell culture are listed in Supplementary Table S7. Recombinant proteins used for cell culture are listed in Supplementary Table S8.

## Hematoxylin and eosin staining on murine paraffin-embedded lung and gut sections

Lung lobes and segments of the gut were excised, fixed in 10% formalin-PBS solution, dehydrated, and embedded in paraffin. Sections of lung and gut tissue, 5 micrometers thick, were cut from paraffin blocks and stained with hematoxylin and eosin to visualize lung tumors and gut inflammation. The stained sections were scanned using a digital slide scanner (Scan 150, 3D Histech Ltd), and images were analyzed with CaseViewer software (Version 2.0, 3D Histech Ltd, RRID:SCR_017654). Hematoxylin and eosin staining was carried out in collaboration with the Institute of Pathology at the University Hospital Erlangen.

## Tumor load detection

Tumor load was quantified using QuPath v0.5.0 (RRID:SCR_018257). It was defined as the ratio of tumor area to total lung tissue observed in histological sections. Total lung tissue was identified with the simple tissue detection tool. For validation, lung sections were additionally examined histologically by a pathologist from the Institute of Pathology at the University Hospital Erlangen in a blinded study.

## Bioluminescence cytotoxic detection assay

For cytotoxic analysis of total lung cell supernatant, LL/2-luc-M38 cells (RRID:CVCL_5J40) were seeded at approximately $7 \times 10^3$ cells per well in 96-well white-walled plates (Thermo Fisher Scientific, Waltham, MA, USA, Cat#165306) and cultured for 24 h at 37 °C in 5% CO2. After 24 h, the culture supernatant was replaced: wells were washed with 100 μl PBS after discarding the medium. Supernatant from 24-hour lung cell cultures prepared from murine samples in our laboratory was diluted 1:3 with DMEM Medium (Gibco™ DMEM, high glucose, Thermo Fisher Scientific, cat# 11965092) supplemented with 10% heat-inactivated fetal bovine serum (FCS) (Sigma-Aldrich, Cat# S0615), 1% Penicillin + Streptomycin (Pen/Strep) (anprotec, Cat# AC-AB-0024), and 1% L-Glutamine (2 mM L-Glutamine, anprotec, Cat# AC-AS-0001). LL/2-luc-M38 cells were cultured in this conditioned medium for 24 h in quadruplicates. Afterward, the medium was replaced with 100 μl luciferin solution (1500 μg/ml, Promega, Cat#P1043) diluted 1:10 in PBS, and cells were incubated for 25 min at 37 °C in 5% CO2. Bioluminescence was measured at room temperature using a Centro XS3 LB 960 plate-reading luminometer (Berthold Technologies, Bad Wildbad, Germany) with a counting time of 0.5 seconds. The mean values of quadruplicates were then calculated.

## Flow cytometric analysis of murine serum cytokine levels and murine total lung cell supernatant cytokine levels

On the day of sacrifice, blood was collected from mice in MiniCollect Tubes (Greiner Bio One, Cat#450536) and centrifuged (Thermo Fisher Scientific, Megafuge™ 16 Universal-Zentrifuge, Cat# 75004270) for 10 min at 3000 × g and room temperature to separate serum from other blood components. For cell culture, $1 \times 10^6$ murine total lung cells were seeded in 48-well plates with Gibco™ RPMI 1640 Medium, Thermo Fisher Scientific (cat# 21875091), supplemented with 10% heat-inactivated fetal bovine serum (FCS) (Sigma-Aldrich, Cat# S0615), 1% Penicillin + Streptomycin (Pen/Strep) (anprotec, Cat# AC-AB-0024), and 1% L-Glutamine (2 mM L-Glutamine, anprotec, Cat# AC-AS-0001). Cells were either left unstimulated or stimulated with 5000 ng/ml plate-bound anti-CD3 (1 mg/ml, BD Bioscience, Cat#555329, RRID: AB_395736) and 2000 ng/ml anti-CD28 (1 mg/ml, BD Bioscience, Cat#555725, RRID: AB_396068). Cultures were maintained for 24 h at 37 °C in 5% CO2, after which supernatants were collected. Both serum and cell culture supernatants were analyzed using the LEGENDplex™ MU Th Cytokine Panel (12-plex) w/ VbP V03 kit (BioLegend, Cat#741044), and beads were acquired on the BD FACSymphony™ A1 (RRID:SCR_024830).

## Flow cytometric analysis of human and murine cells

Flow cytometric analysis was carried out by seeding 1,000,000 cells in 96-well U-bottom plates, followed by centrifugation (Thermo Fisher Scientific, Megafuge™ 16 Universal-Zentrifuge, Cat# 75004270). All centrifugation steps were performed for 1 min at 2000 rpm and 4 °C. After discarding the supernatant, cells were pre-incubated with anti-CD16/CD32 (1:100, BD Biosciences, Cat# 553142, RRID: AB_394657) mAb for at least 10 min at room temperature in the dark to block non-specific immunoglobulin binding to Fc receptors. Cells were washed with 150 μl FACS buffer (PBS + 2% FCS, Sigma-Aldrich, Cat# S0615), centrifuged, and subsequently stained with 50 μl of a surface antibody mix for 20 min at 4 °C in the dark. After washing again with 150 μl FACS buffer and centrifugation, intracellular staining was performed by resuspending cells in 100 μl FoxP3 Fixation/Permeabilization reagent (Thermo Fisher Scientific, Cat# 00-5523-00) and incubating for 30 min at room temperature in the dark. Cells were washed twice with 200 μl permeabilization buffer (1 part Thermo Fisher Scientific Permeabilization Buffer (10X), Cat# 00-8333-56, and 9 parts Millipore-H2O), stained with 50 μl intracellular antibody mix for 30 min at room temperature in the dark, and washed twice more with 200 μl permeabilization buffer. Finally, cells were resuspended in 100 μl FACS buffer and analyzed on a BD FACSymphony™ A1 flow cytometer (BD Biosciences, Heidelberg, RRID:SCR_024830).

The antibodies used are listed in Supplementary Table S9: anti-murine-CD3 (1:150 APC-Cy7, BioLegend, Cat#100329, RRID: AB_1877171), anti-murine-CD4 (1:200 BV605, BioLegend, Cat#100451, RRID: AB_2564591), anti-murine-CD8 (1:150, BV711, BioLegend, Cat#100747, RRID: AB_11219594), anti-murine-CD194 (1:100, BV421, BioLegend, Cat#131217, RRID: AB_10933253), anti-murine-Ly-6C (1:100, PE-Cy7, BD Biosciences, Cat# 560593, RRID:AB_1727557), anti-murine-GR-1 (1:100, PE, Miltenyi Biotec, Cat# 130-102-426, RRID:AB_2659861), anti-murine-Ly-6G (1:100, APC, Miltenyi Biotec, Cat# 130-093-140, RRID:AB_1036094), anti-murine-CD11b (1:150, APC-Fire, BioLegend, Cat# 101261, RRID:AB_2572121), Zombie Aqua™ Fixable Viability Kit (1:500, BV510, BioLegend, Cat# 423101), anti-murine-CD3e (1:100, PE-Cy7, BD Biosciences, Cat# 552774, RRID:AB_394460), anti-murine-NKG2D (1:100, PE, Thermo Fisher Scientific, Cat# 12-5882-81, RRID:AB_465995), anti-murine-CD49b (1:100, FITC, BD Biosciences, Cat# 553857, RRID:AB_395093), anti-murine-CD335 (1:100, PerCP/Cyanine5.5, BioLegend, Cat# 137609, RRID:AB_10642684), anti-murine-CD45 (1:200, Alexa Fluor(R), BioLegend, Cat# 157615, RRID:AB_2890719), anti-murine-RT1B (1:100, APC, Miltenyi Biotec, Cat# 130-108-708, RRID:AB_2653375), and Fixable Viability Stain 575 V (1:100, BV605, BD Biosciences, Cat# 565694, RRID:AB_2869702). Data were analyzed with Kaluza Analysis 2.1 software (Beckman Coulter, RRID:SCR_016182).

## Quantitative real-time PCR (qPCR)

Total RNA was extracted from frozen tissue samples using Qiazol Lysis® Reagent (QIAGEN, Cat#79306) according to the manufacturer's instructions. 1 μg of the resulting RNA was reverse transcribed into copy DNA (cDNA) via the RevertAid™ First Strand cDNA Synthesis Kit (ThermoFisher Scientific, Cat#K1622) according to the manufacturer's protocol. Each qPCR reaction mix contained 15 ng of cDNA, 300 nM transcript-specific forward and reverse primer iTaq Universal SYBR Green Supermix (Bio-Rad Laboratories, Cat# 1725124) in a total volume of 20 μl. qPCR primers were purchased from Eurofins-MWG-Operon, (Ebersberg, Germany). Primer sequences for human and murine qPCR analysis are shown in the Supplementary Tables S10, S11 as well as the housekeeping genes for human and murine. Reactions were performed for 50 cycles with an initial activation for 2 min at 98 °C, denaturation for 5 min at 95 °C and hybridization and elongation for 10 min at 60 °C. qPCR reactions were performed using the CFX-96 Real-Time PCR Detection System (BIO-RAD, Munich, Germany) and analyzed via the CFX Manager Software. The relative expression level of specific transcripts was calculated using the relative quantification 2-ΔΔCT method with respect to the internal standard ribosomal protein L30 (RPL30) and data with ΔCq values higher than 35 were excluded.

## Wound healing assay

For the analysis of A549 (RRID: CVCL_A549) cell migration, 2 ml with $5 \times 10^5$ cells were seeded into a 6-well plate (Thermo Fisher Scientific, Greiner Bio-One CELLSTAR 6-well cell culture plate, Cat# 10536952). Cells were cultured to adhere for 24 h in Gibco™ RPMI 1640 Medium, Thermo Fisher Scientific (cat# 21875091) supplemented with 10% of heat-inactivated fetal bovine serum (FCS) (Sigma-Aldrich, Cat# S0615), 1% Penicillin + Streptomycin (Pen/Strep)(anprotec, Cat# AC-AB-0024) and 1% L-Glutamine (2 mM L-Glutamine, anprotec, Cat# AC-AS-0001) and were grown to 95% confluence. Cells were incubated at 37 °C, 5% $CO_2$ and 96% RH. A 200 μl pipet tip was used to scratch the cell monolayer multiple times, 1 hour prior to the scratch 5 μg/ml Mitomycin C (Sigma-Aldrich, Cat#M4287) was applied to remove any proliferation effects. Cells were washed 3 times with 1 ml PBS followed by detection of cell migration. New medium was added and left either unstimulated or stimulated with 100 μM alpha-linolenic acid (Sigma-Aldrich, Cat#L2376). Cell migration was detected over a period of 48 h using the Keyence BZ-X800 microscope. The extent of healing was defined by the ratio of the difference between the resulting wound areas compared with the original wound area. Therefore, ImageJ2 was used.

## Statistics and reproducibility

Differences were evaluated using GraphPad Prism 10 (RRID:SCR_002798) for significance (*P,0.05; **P,0.01; ***P,0.001; ****P,0.0001). Data were imported in column statistics and were then analyzed for normal distribution with Shapiro-Wilk test and D'Agostino & Pearson test. Statistical test was chosen depending on normal distribution or non-normal distribution. For normal distribution one-way ANOVA multiple comparisons was used. For non-normal distribution or n < 5 the Kruskal-Wallis test was used. For two independent columns unpaired t-test was used and analyzed for significance with Mann-Whitney test. For two dependent columns paired t-test was used for normal distributed values, Wilcoxon test was used for non-normal distribution. Correlations were examined by importing data in XY-tables and diagramed with linear regression curve. The two-tailed Pearson correlation analysis was performed to get the r and p value. Data are given as mean values ± SEM. Graphs were created with GraphPad Prism 10 (Windows).

For the patient cohort, each individual patient sample represents a biological replicate. For mouse studies, n indicates the number of individual animals per group, as specified in the figure legends. All in vivo experiments were independently repeated at least three times to confirm reproducibility. For in vitro assays (cell culture, qPCR, flow cytometry), biological replicates denote independent experiments performed on different days or using separate cultures, while technical replicates refer to repeated measurements of the same sample (e.g., qPCR reactions run in triplicate, flow cytometry runs with technical repeats). All key findings were independently confirmed, with the number of biological samples or animals reported throughout the manuscript.

## Results
### Upregulated metabolic pathway of potentially tumor driving oxysterol 27-hydroxycholesterol in the tumoral region of patients with NSCLC

In this study, we sought to characterize diet-derived lipids' influence on alteration of the TME in patients with non-small cell lung cancer (NSCLC). Key clinical characteristics of the patient cohort are summarized in Table 1.

The study population was gender-balanced, with 51.5% female and 48.5% male participants and the mean age was 65.39 years. Anthropometric data revealed an average body mass index (BMI) of 26.71, classifying the group as slightly overweight. A mean waist circumference of 103.4 cm (data available for 15 patients) suggests central adiposity in a subset of patients. Tumor-related parameters showed a mean tumor diameter of 2.91 cm, consistent with early to intermediate disease stages at diagnosis or surgery.

**Table 2 | Clinical data of the female patients analyzed in this study**

| Characteristics female participants | Mean | | SEM | SD | N |
|---|---|---|---|---|---|
| Age | 66.29 | ± | 2.44 | 10.07 | 17 |
| Body weight [kg] | 69.47 | ± | 4.45 | 18.35 | 17 |
| Body height [m] | 1.61 | ± | 0.01 | 0.04 | 17 |
| BMI = Weight/Height² | 26.86 | ± | 1.59 | 6.55 | 17 |
| Waist circumference [cm] | 106.14 | ± | 5.47 | 22.57 | 7 |
| Tumor Diameter [cm] | 2.86 | ± | 0.40 | 1.66 | 17 |
| Smoking [pack years] | 33.67 | ± | 6.54 | 26.96 | 16 |
| Albumin presurgical [g/l] | 42.24 | ± | 1.06 | 4.39 | 16 |
| CRP presurgical [mg/l] | 7.39 | ± | 3.48 | 14.36 | 16 |
| CRP postsurgical [mg/dl] | 89.74 | ± | 14.46 | 59.64 | 17 |
| Lymphocytes absolute presurgical [x10³/μl] | 1.93 | ± | 0.17 | 0.68 | 17 |
| Lymphocytes before surgery [%] | 25.00 | ± | 1.88 | 7.73 | 17 |
| FEV1 [%] | 87.33 | ± | 4.06 | 16.72 | 16 |
| FVC [%] | 98.74 | ± | 1.94 | 8.00 | 14 |
| Glucose [mg/dl] | 107.40 | ± | 10.44 | 43.03 | 15 |
| Total cholesterol presurgical [mg/dl] | 202.54 | ± | 10.07 | 41.54 | 13 |
| HDL cholesterol presurgical [mg/dl] | 58.62 | ± | 3.99 | 16.47 | 13 |
| LDL cholesterol presurgical [mg/dl] | 118.92 | ± | 7.31 | 30.14 | 13 |
| Triglycerides presurgical [mg/dl] | 111.23 | ± | 9.34 | 38.51 | 13 |
| Lipoprotein A presurgical [mg/dl] | 16.65 | ± | 4.13 | 17.01 | 13 |

The cohort demonstrated a substantial history of tobacco exposure, with an average of 37.52 pack-years among 29 patients. Markers of systemic inflammation were elevated, as reflected by presurgical C-reactive protein (CRP) levels (mean: 13.43 mg/L) and a marked postoperative increase (107.7 mg/dL), consistent with surgical trauma and inflammatory response. Albumin levels remained within the normal range (mean: 41.79 g/L), indicating preserved nutritional status in most patients. Immunological profiling revealed a mean absolute lymphocyte count of 1.86 ×10³/μl and a relative lymphocyte percentage of 24.61%, both within expected physiological ranges. Pulmonary function tests indicated largely preserved respiratory capacity, with a mean FEV1 of 84.3% and FVC of 94.36% of predicted values. Metabolic assessment showed a mean fasting glucose level of 106.18 mg/dL, marginally elevated above the normoglycemic range. Lipid profiling revealed total cholesterol levels averaging 179.19 mg/dL, with HDL at 50.00 mg/dL, LDL at 105.59 mg/dL, and triglycerides at 120.96 mg/dL. Lipoprotein A levels averaged 17.20 mg/dL, without signs of marked dyslipidemia in the cohort.

When comparing males and females in our study, we first separated Table 1 that included all participants regardless of gender into female and male groups, as displayed in Tables 2 and 3, and upon further analysis, several differences emerged in their physical and clinical measurements: males were generally taller and heavier than females, with average heights of 1.78 m compared to 1.61 m, and weights around 85 kg versus 69 kg, respectively. Despite these differences, both groups had similar BMI values, roughly 26.5 to 26.9. Inflammation markers stood out, with males showing significantly higher C-reactive protein (CRP) levels before and after surgery. This was consistent with their reported higher lifetime tobacco exposure, measured in pack years. Lipid profiles also differed. Females had higher total cholesterol, high-density lipoprotein (HDL), and low-density lipoprotein (LDL) levels, while males had somewhat elevated triglycerides. Levels of lipoprotein A were similar between the sexes. Tumor size was comparable but slightly larger in males. Immune cell counts, measured by lymphocyte numbers and percentages, did not differ significantly. Albumin levels were marginally higher in females, and blood glucose concentrations were similar across both groups.

**Table 3 | Clinical data of the male patients analyzed in this study**

| Characteristics male participants | Mean | | SEM | SD | N |
|---|---|---|---|---|---|
| Age | 64.44 | ± | 2.27 | 9.08 | 16 |
| Body weight [kg] | 84.88 | ± | 4.48 | 17.92 | 16 |
| Body height [m] | 1.78 | ± | 0.02 | 0.07 | 16 |
| BMI = Weight/Height² | 26.55 | ± | 1.13 | 4.53 | 16 |
| Waist circumference [cm] | 101.00 | ± | 2.43 | 9.71 | 8 |
| Tumor Diameter [cm] | 3.14 | ± | 0.50 | 2.00 | 16 |
| Smoking [pack years] | 41.64 | ± | 7.89 | 31.58 | 15 |
| Albumin presurgical [g/l] | 41.18 | ± | 0.78 | 3.12 | 12 |
| CRP presurgical [mg/l] | 21.88 | ± | 9.99 | 39.96 | 13 |
| CRP postsurgical [mg/dl] | 128.05 | ± | 25.03 | 100.12 | 16 |
| Lymphocytes absolute presurgical [x10³/µl] | 1.79 | ± | 0.18 | 0.70 | 15 |
| Lymphocytes before surgery [%] | 24.16 | ± | 2.80 | 11.18 | 15 |
| FEV1 [%] | 80.84 | ± | 5.61 | 22.44 | 14 |
| FVC [%] | 89.65 | ± | 4.93 | 19.71 | 13 |
| Glucose [mg/dl] | 104.77 | ± | 2.65 | 10.60 | 13 |
| Total cholesterol presurgical [mg/dl] | 157.50 | ± | 6.34 | 25.35 | 14 |
| HDL cholesterol presurgical [mg/dl] | 42.00 | ± | 4.82 | 19.29 | 14 |
| LDL cholesterol presurgical [mg/dl] | 93.21 | ± | 5.15 | 20.60 | 14 |
| Triglycerides presurgical [mg/dl] | 130.00 | ± | 10.79 | 43.15 | 14 |
| Lipoprotein A presurgical [mg/dl] | 17.71 | ± | 5.79 | 23.15 | 14 |

In summary, males showed more signs of inflammation and lower lung function, likely influenced by heavier smoking history. Females displayed a more favorable HDL cholesterol profile and better lung capacity, despite higher overall cholesterol levels.

To further investigate a previously reported potential protumorigenic metabolic pathway of cholesterol[11], we performed real-time qPCR on post-surgery lung samples collected from the tumoral region (TU: solid tumor tissue) and the tumor-free control region (CTR: >5 cm away from the solid tumor) in NSCLC (*see Materials and Methods*, Supplementary Fig. S1A). Moreover, we wanted to investigate the effect of cholesterol on this area affecting anti-tumor immune responses. The clinical data of these patients, whose post-surgery lung samples were analyzed in this study, are reported in Tables 1 and S1–S4, respectively.

The CYP27A1 gene encodes a cytochrome P450 oxidase that is also known as sterol 27-hydroxylase. It catalyzes hydroxylation of cholesterol to 27-hydroxycholesterol (27-HC). Figure 1A shows that CYP27A1 mRNA expression was significantly higher in tumoral regions (mean fold change = 2.15 ± 0.34, n = 27) compared to matched control tissue (n = 25, P = 0.0163). At the same time, CYP7B1 expression (Fig. 1B) was significantly downregulated in tumoral areas (n = 28) versus control (n = 33), with a mean decrease of ~45% (P = 0.0006), suggesting impaired oxysterol degradation and potential accumulation of 27-HC. Previous studies have found 27-hydroxycholesterol accumulating in CYP7B1 knockout models[11], suggesting accumulation of this metabolite in our cohorts' tumoral area in the lung tissue samples. Interestingly, here we found an inverse correlation between the tumor diameter and CYP7B1 gene expression (Supplementary Fig. S1B). To get a better understanding of how 27-hydroxycholesterol influences the TME we next analyzed downstream metabolic pathways of 27-HC. It has been reported that 27-HC, among others, induces phosphorylation of NFkB p65, AKT and peptidylprolyl isomerase B (PPIB)[11,19–22]. Therefore, we measured gene expression of **PPIB**, **NFkB p65** and **AKT** in the tumoral and control area of our cohorts' lung tissue samples. Gene expression of downstream mediators PPIB, NFkB p65, and AKT (Fig. 1C–E) were also significantly increased in the tumor tissue compared to control. Specifically, PPIB

(n = 7 tumor, n = 9 control, P = 0.0312), NFkB p65 (n = 28 tumor, n = 26 control, P < 0.0001), and AKT (n = 25 tumor, n = 24 control, P = 0.0277) showed consistent elevation, reinforcing a pro-tumoral signaling profile. To investigate the influence of 27-HC-driven induction of these key-players in NSCLC progression, we further analyzed molecular actors of EMT in lung cancer, SNAIL and Vimentin[23,24]. EMT-related genes SNAIL and Vimentin were upregulated in tumoral tissue (Fig. 1F, G, respectively), with SNAIL mRNA showing significant increase (P = 0.0312, n = 7 tumor vs. n = 8 control), as well as Vimentin (P = 0.0413, n = 15 tumor vs. n = 16 control).

To determine whether or not cholesterol accumulation within the cancer cell through ACAT upregulation[2] is connected to metabolization of cholesterol to 27-HC via CYP27A1[11] and downstream EMT induction, we correlated ACAT expression with CYP27A1 expression. Linear regression analyses revealed strong correlations between ACAT1 and CYP27A1 expression in both tumoral (n = 27, R² = 0.30, P = 0.0052) and control (n = 24, R² = 0.20, P = 0.0334) regions (Fig. 1H), suggesting ACAT1-driven cholesterol influx fuels oxysterol synthesis. This correlation held in both tumoral and control region lung tissue of patients with NSCLC, indicating a possible relationship between intracellular cholesterol accumulation, metabolization and induction of proliferative, immune evasive and metastatic pathways in cancer cells. Similarly, significant positive correlations were found between NFkB p65 and CYP27A1 (Fig. 1I; n = 26 tumor, R² = 0.41, P = 0.0005), AKT and NFkB p65 (Fig. 1J; n = 24 tumor, R² = 0.76, P < 0.0001), and Vimentin and NFkB p65 (Fig. 1K; n = 15 tumor, R² = 0.61, P = 0.0006). Moreover, as displayed in Supplementary Fig. S1C, CYP27A1 mRNA expression level correlated positively with mRNA expression levels of Vimentin, suggesting cholesterol metabolism might be linked to Vimentin mediated EMT.

In order to understand the role of dietary lipids in cell migration, we started a series of cell scratch experiments with the human adenocarcinoma cell line A549 (*See Materials and Methods*) (Fig. 1L). Cells were grown in RPMI containing 10% FCS, 1% P/S and 1% L-Glu and were grown to 95% confluence. The cells were then scratched and either treated with 100 µM of α-linolenic acid, an ω−3 PUFA, or left untreated. Cell migration was monitored over a period of 48 h, capturing the region of interest six times per hour. The extent of healing was defined by the ratio of the difference between the resulting wound areas compared with the original wound area. Cell migration assays (Fig. 1M) demonstrated that treatment with 100 µM α-linolenic acid (ω−3 PUFA) significantly reduced wound closure in A549 cells over 48 h compared to untreated control (P = 0.0047, n = 8/group) (Fig. 1M and Supplementary Fig. S2A, B).

**Cholesterol-containing, but not ω-3 poly-unsaturated fatty acid containing diets increased tumor load in tumor bearing mice**

To understand the effects of dietary fatty acids on LUAD development in vivo, we used an established murine model of lung cancer. Therefore, LUAD (LL/2-Luc-M38) cells were injected intravenously and mice were fed a high fat cholesterol rich arteriosclerosis diet, a cholesterol free ω-3 PUFA-enriched diet and a standard diet (Fig. 2A and Supplementary Tables S5, S6). The diets had different compositions: the standard diet (SD) contained 43% carbohydrates, 19% proteins and 4% fat, whereas the high fat cholesterol rich arteriosclerosis diet (AD) contained 45% carbohydrate, 17% protein and 20% fat and the ω-3 PUFA-enriched diet (OD) consisted of 20.8% fat, 17.6% protein and 25.8% carbohydrates (Fig. 2B, C). The different diets were started eleven days before the injection of LUAD cells. Diet was changed for the intervention diet group one more time three days before tumor cell inoculation. After tumor cell injection, mice were fed consistently for 17 days while their body weight development was monitored. As shown in Fig. 2A, dietary intervention preceded tumor inoculation by 11 days. Mice fed the arteriosclerosis diet (AD) showed increased weight gain post-injection compared to other diet groups, although differences were not statistically significant (Fig. 2D). On day 33, after ending the experiment, the lungs of the mice were removed, and lung cells were isolated, cultured and

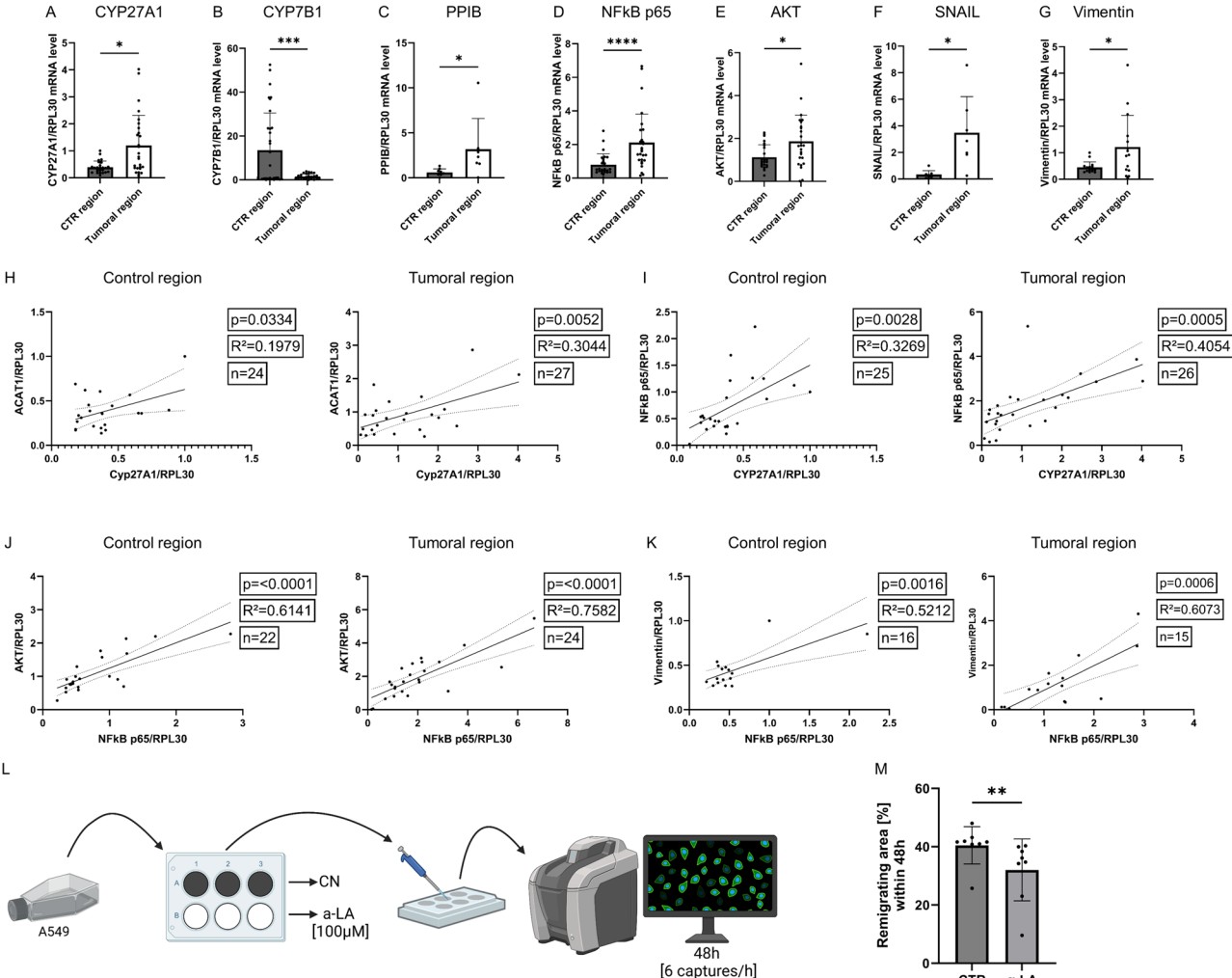

**Fig. 1 | Overexpression of CYP27A1 and downregulation of CYP7B1 in LUAD cells of patients lead to 27-hydroxy-cholesterol accumulation and activated pathways that drive progression and metastasis through NFkB p65, AKT, SNAIL, and Vimentin.** RNA was isolated from the lung tumoral (**TU**) and control (**CTR**) region of patients with LUAD. qPCR was performed and mRNA expression level was related to the housekeeping gene *RPL30*. **A** *CYP27A1* mRNA expression in control (n = 25) and tumoral (n = 27) region. P (CTR *vs.* TU) = 0.0163. **B** *CYP7B1* RNA expression in control (n = 33) and tumoral (n = 28) region. P (CTR *vs.* TU) = 0.0006. **C** *PPIB* RNA expression in control (n = 9) and tumoral (n = 7) region. P (CTR *vs.* TU) = 0.0312. **D** *NFkB p65* RNA expression in control (n = 26) and tumoral (n = 28) region. P (CTR *vs.* TU) = < 0.0001. **E** *AKT* RNA expression in control (n = 24) and tumoral (n = 25) region. P (CTR *vs.* TU) = 0.0277. **F** *SNAIL* RNA expression in control (n = 8) and tumoral (n = 7) region. P (CTR *vs.* TU) = 0.0312. **G** *Vimentin* RNA expression in control (n = 16) and tumoral (n = 15) region. P (CTR *vs.* TU) = 0.0413. **H** Correlation of relative *ACAT1/RPL30* mRNA expression level measured in the control/tumoral region correlated with relative *CYP27A1/RPL30* mRNA expression level measured in control/tumoral region of patients with LUAD; CTR (n = 24, p = 0.0334, $R^2$ = 0.1979), TU (n = 27, p = 0.0052, $R^2$ = 0.3044). **I** Correlation of relative *NFkB p65/RPL30* mRNA expression level measured in the control/tumoral region correlated with relative *CYP27A1/RPL30* mRNA expression level measured in control/tumoral region of patients with LUAD; CTR (n = 25, p = 0.0028, $R^2$ = 0.3269), TU (n = 26, p = 0.0005, $R^2$ = 0.4054). **J** Correlation of relative *AKT/RPL30* mRNA expression level measured in the control/tumoral region correlated with relative *NFkB p65/RPL30* mRNA expression level measured in control/tumoral region of patients with LUAD; CTR (n = 22, p = < 0.0001, $R^2$ = 0.6141), TU (n = 24, p = < 0.0001, $R^2$ = 0.7582). **K** Correlation of relative *Vimentin/RPL30* mRNA expression level measured in the control/tumoral region correlated with relative *NFkB p65/RPL30* mRNA expression level measured in control/tumoral region of patients with LUAD; CTR (n = 16, p = 0.0016, $R^2$ = 0.5212), TU (n = 15, p = 0.0006, $R^2$ = 0.6073). **L** Experimental setup of wound healing assay. [*Created in BioRender. Harre, P. (2025)* https://BioRender.com/j76f918]. **M** % of remigrated area 48 h after scratching confluent cell surface. Extent of remigration was defined by the ratio of the difference between the remaining wound areas compared with the original wound area. Conditions used were control (CTR; n = 8) and α-linolenic acid (α-LA; n = 8) stimulated. P (CTR *vs.* α-LA) = 0.0047. Correlations (**H–K**) are shown by using simple linear regression. The two-tailed Pearson correlation analysis was performed to get the r and *p* value for (**H–K**). Wilcoxon test was performed for (**A, C–G**). Paired t-test was performed for (**B**). Mann-Whitney test was used for (**M**). All statistical tests were two-sided. No adjustments for multiple comparisons were applied. Data are shown as mean values ± s.e.m.; *P < 0.05; **P < 0.01 ***P < 0.001; ****P < 0.0001.

analyzed. In vivo tumor burden was quantified using bioluminescence imaging (Fig. 3A, B). Mice on the AD diet displayed significantly higher average radiance (n = 9) compared to those on the OD diet (n = 9, P = 0.0015) and SD (n = 8, P = 0.0139). To verify the in vivo imaging finding showing a significant increase of tumor load in the group fed with high fat, high cholesterol diet (arteriosclerosis diet, AD), sections of the lung were analyzed histologically at the end of the experiment. Therefore, lung sections

were stained with hematoxylin and eosin (HE) and analyzed in a blinded study by an expert pathologist. Histologic quantification of tumor area (Fig. 3C) corroborated imaging results: mice fed OD (n = 39) or ID (n = 36) had significantly reduced tumor burden versus SD (n = 36, P = 0.0303) and AD (n = 36, P = 0.0184). Representative lung sections are shown in Fig. 3D. Furthermore, we noticed decreased inflammation in the gut of mice that were fed ω-3 rich diets (OD, ID) (Supplementary Fig. S3A). Next, we

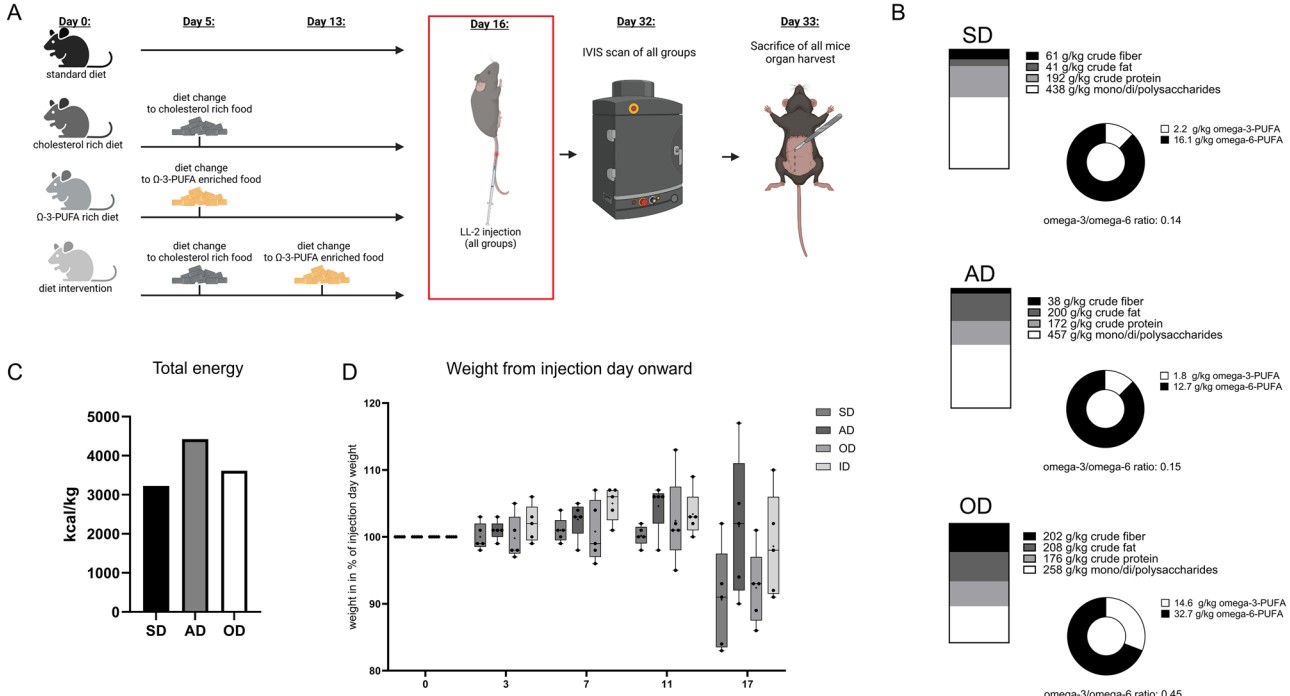

**Fig. 2 | Experimental set up and composition overview of diets used in the in vivo experiments. A** Mice were fed with their corresponding diet and had diet changes at indicated time points. At 8 weeks of age, mice were intravenously injected with $5 \times 10^5$ LL/2-luc-M38 cells in the tail vain. Tumor growth was detected using in vivo imaging system (IVIS) one day before sacrificing the mice. [*Created in BioRender. Harre, P. (2025)* https://BioRender.com/q87j248]. **B** Composition of food fed to the mice [SD: food (Altromin, 1324); AD: modified arteriosclerosis diet (Altromin, C-1061-1), OD: low cholesterol rapeseed oil enriched diet (Altromin, C-1060)]. **C** Total energy in kcal/

kg in diets that were fed to the mice: SD (standard diet) = 3226 kcal/k; AD (high cholesterol arteriosclerosis diet) = 4423 kcal/kg, OD (ω-3 PUFA rich diet) = 3613 kcal/kg. **D** Weight development of mice throughout the course of the experiment in [%] related to their weight on the day of tumor cell inoculation (SD: standard diet, n = 5; AD: high cholesterol arteriosclerosis diet, n = 5; OD: ω-3 PUFA rich diet, n = 5; ID: intervention diet, n = 5). Data are presented as box-and-whisker plots. Boxes indicate the interquartile range (25th–75th percentile), horizontal lines mark the median, whiskers represent data range, and dots indicate individual animals.

investigated serum analytes of the mice focusing on serum glucose and lipids. In a previous study, we found high blood glucose levels associated with worse survival rates in patients with LUAD[25]. Serum glucose analysis (Fig. 3E) revealed elevated levels in cholesterol-containing diet groups (n = 32) versus cholesterol-free groups (n = 20; P = 0.0262). Lipid profiling confirmed diet-induced differences: total serum cholesterol (Fig. 3F) was highest in AD-fed mice (n = 12, P = 0.0084 vs. SD), while LDL (Fig. 3H) was significantly elevated in AD versus SD (P = 0.0338) and ID (P = 0.0420), indicating a shift toward a lung cancer progression and metastasis favoring blood lipid profile[26,27]. Simultaneously, HDL (Fig. 3G) was increased in OD and ID compared to AD (P = 0.0323 and P = 0.0032, respectively), showing a converse shift to a lipid profile negatively associated with lung cancer risk and progression[28,29].

Taken together, our data indicate that fatty acids in diet play a key role in lung cancer development and progression. Mice with increased serum total cholesterol and LDL developed significantly more tumor mass throughout our experiments, while serum HDL seemed to exert a protective effect.

### Dietary fatty acids influence serum TNFα and IFNγ cytokine levels

To further investigate serum cytokine profile of the different diet groups, we analyzed murine serum samples aiming at measuring TNFα and IFNγ serum levels. Tumor necrosis factor (TNFα) is a cytokine known for decades and highly associated with inflammation-induced cancer. As shown in Fig. 4A, serum TNFα levels were elevated in AD (mean: 71.6 pg/mL, n = 17) and OD (65.4 pg/mL, n = 14) groups, with significantly lower levels in ID mice (39.2 pg/mL, n = 16; P = 0.0061 vs. OD, P = 0.0177 vs. AD). IFNγ is known to be one of the most important cytokines in anti-tumor immunity and upregulation of the IFNγ gene expression is the most prominent feature of modern immune checkpoint inhibition therapies[30]. Serum IFNγ (Fig. 4B)

was highest in OD (42.3 pg/mL, n = 13) and showed a positive trend compared to ID (31.1 pg/mL, n = 16; P = 0.0873), consistent with enhanced antitumoral immunity. Moreover, there were no significant differences in serum levels of the cytokines IL-5, IL-4 and IL-6 (Supplementary Fig. S3B–D).

### Dietary ω-3 rich PUFAs averted T cell mediated secretion of cancer driving cytokines in the TME

To get a better understanding of how dietary lipids can influence the TME, we first investigated T cell mediated cytokine secretion by isolating murine total lung cells from mice fed with our corresponding diet (*see Materials and Methods*). Next, we cultured these cells with or without anti-CD3 and anti-CD28 antibodies to induce T cell activation in vitro. After 24 h of cell culture, we harvested the lung cells and collected their supernatants which was then analyzed (Fig. 4C and Supplementary Fig. S4A). In lung cell culture supernatants from anti-CD3/anti-CD28-stimulated cells (Fig. 4C), IL-4 (Fig. 4D) was significantly lower in OD (n = 8) compared to SD (n = 8, P = 0.0377) and AD (n = 13, P = 0.0004), and also lower than ID (n = 13, P = 0.0122), indicating induction of IL-4/STAT6 axis in mice fed with standard or cholesterol rich food when compared to mice that had been administered ω-3 rich PUFA food. Another promotor of EMT and critical factor in the development and metastasis of NSCLC is IL-6 through numerous pathways. Mediation of chemoresistance via NFkB pathway activation as well as macrophage mediated EMT induction have been reported among others and circulating IL-6 levels have been used as a prognostic marker since its expression is associated with poor prognosis[31–35]. IL-6 levels (Fig. 4E) were reduced in OD (n = 4) and ID (n = 5) versus SD (n = 4; P = 0.0286 for OD vs. SD; P = 0.0042 for ID vs. SD). Furthermore, we investigated IL-9 levels in the lung cell supernatants as we have previously discovered

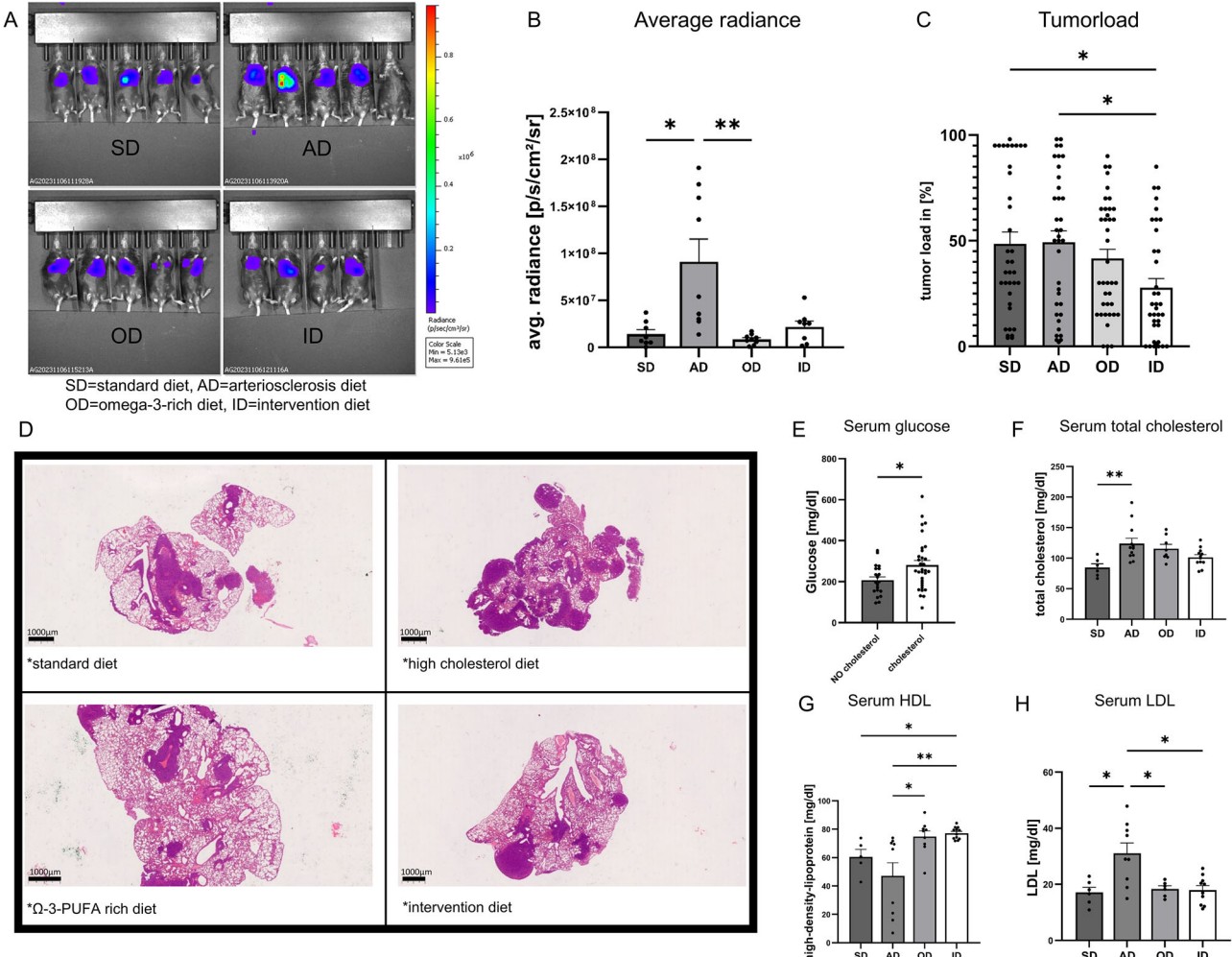

**Fig. 3 | Cholesterol promotes tumor growth and progression in vivo.** Analysis of relevant serum lipids: **A** Representative in vivo imaging system sequence image of tumor bearing mice of all groups. **B** Quantitative analysis of in vivo imaging system calculated as average radiance [p/s/cm²/sr] (SD, standard diet n = 8; AD, arteriosclerosis diet n = 9; OD, ω-3 PUFA-rich diet n = 9; ID, intervention diet n = 8). P (SD vs. AD) = 0.0139; P (AD vs. OD) = 0.0015. **C** Semiquantitative analysis of tumor load performed on H&E stained lung tissue slices of three independent experiments, three slides were analyzed for each animal (SD, standard diet n = 36; AD, arteriosclerosis diet n = 36; OD, ω-3 PUFA-rich diet n = 39; ID, intervention diet n = 36). P (SD vs. ID) = 0.0303; P (AD vs. ID) = 0.0184. **D** Representative picture of HE-stained lung slides from one lobe of the lung of tumor bearing mice. **E** Serum glucose levels [mg/dl] of mice; one group received food without cholesterol (n = 20), the other group received food containing cholesterol (n = 32). P (diet containing no cholesterol vs. diet containing cholesterol) = 0.0262. **F** Serum total cholesterol levels of diet groups (SD, standard diet n = 6; AD, arteriosclerosis diet n = 12; OD, ω-3 PUFA rich diet n = 8; ID, intervention diet n = 12). P (SD vs. AD) = 0,0084. **G** Serum high density lipoprotein (HDL) levels of diet groups (SD, standard diet n = 5; AD, arteriosclerosis diet n = 9; OD, ω-3 PUFA-rich diet n = 9; ID, intervention diet n = 12). P (AD vs. ID) = 0.0032; P (AD vs. OD) = 0.0323; P (SD vs. ID) = 0.0480. **H** serum low density lipoprotein (LDL) levels of diet groups (SD, standard diet n = 6; AD, arteriosclerosis diet n = 9; OD, ω-3 PUFA-rich diet n = 6; ID, intervention diet n = 10). P (SD vs. AD) = 0.0338; P (AD vs. ID) = 0.0420. Mann-Whitney-test was used for (**E**). Kruskal-Wallis test was used for (**B**, **F**, **G**). One-way ANOVA test was used for (**C**, **H**). All statistical tests were two-sided. No adjustments for multiple comparisons were applied. Data are shown as mean values ± s.e.m.; *P,0.05, **P,0.01.

and reported IL-9 producing Tregs mediating tumoral immune evasion in NSCLC and suppression of tumor development as well as reduced IL-10 production in the lung[36]. In this study, IL-9 secretion (Fig. 4F) followed a similar trend with lower levels in OD (n = 4) compared to SD (n = 4, P = 0.0375) and ID vs. SD (P = 0.0331).

IL-10 is regarded as immunosuppressive cytokine. As shown in Fig. 4G, IL-10 was highest in AD (n = 5) and significantly increased compared to OD (n = 4; P = 0.0356), suggesting ω-3 PUFA diets may mitigate IL-10-mediated immune suppression. Next, we sought to investigate the role of IL-17A in this model of lung cancer. Although its role in lung cancer development remains largely unknown, studies have found it to be a promotor of tumor progression in different types of cancer, suggesting an angiogenic role of IL-17A in tumor[37–39]. To connect angiogenesis in NSCLC to dietary lipid intake, we therefore measured IL-17A in lung cell culture supernatants and found IL-17A (Fig. 4H) was significantly higher in AD-fed

mice (n = 8) compared to OD (n = 6; P = 0.0283), consistent with enhanced inflammation, indicating upregulated angiogenesis and immune suppression in mice fed with AD.

In the context of cancer, another tumor progression inducing cytokine is IL-5 which has been shown to induce the secretion of CC-chemokine ligand 22 (CCL22) by eosinophils, thus resulting in the recruitment of immunosuppressive CCR4+ regulatory T cells (**Treg**) to the lung[40]. Cell culture supernatant analysis revealed IL-5 (Fig. 4I) was significantly reduced in OD (n = 4) and ID (n = 5) compared to SD (n = 4; P = 0.0328 and P = 0.0358, respectively).

## IL-5 induced migration of CCR4+ T regulatory cells in the TME

To determine whether or not IL-5 levels in the lung were connected to CCR4+ Treg migration, we measured CD3+ CD4+ CCR4+ cells in murine blood and lung tissue (Supplementary Fig. S4B). Flow

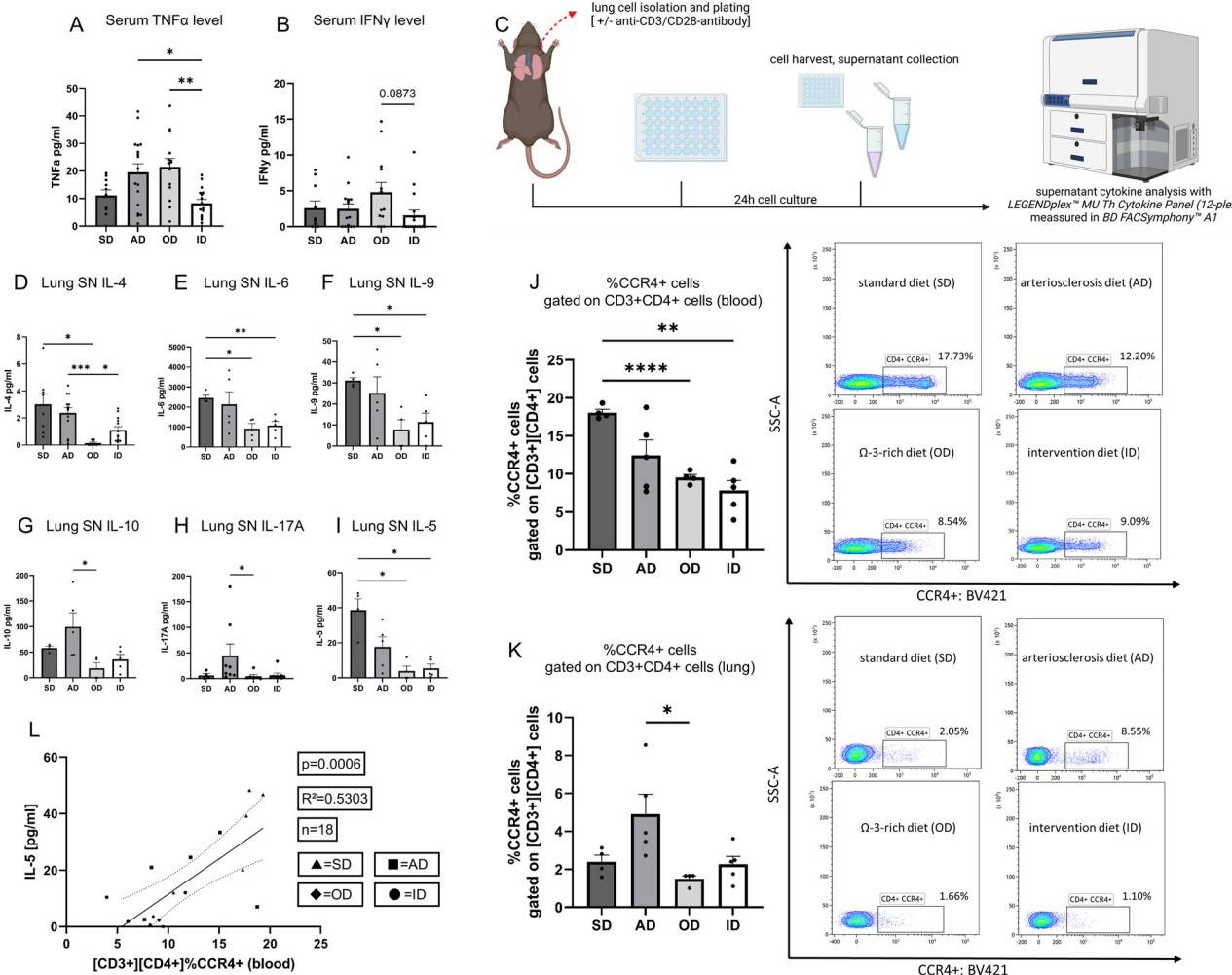

**Fig. 4 | Cytokine analysis revealed induction of EMT and tumor progressing cytokine secretion in murine serum and murine lung cell culture supernatant.** **A, B** Murine serum cytokine levels were measured by *LEGENDplex™ MU Th Cytokine Panel (12-plex)*. **A** Murine serum TNFα level [pg/ml] (SD, standard diet n = 10; AD, arteriosclerosis diet n = 17; OD, ω-3 rich diet n = 14; ID, intervention diet n = 16). P (AD *vs*. ID) = 0.0177; P (OD *vs*. ID) = 0.0061. **B** Murine serum IFNγ level [pg/ml] (SD, standard diet n = 10; AD, arteriosclerosis diet n = 15; OD, ω-3 rich diet n = 13; ID, intervention diet n = 16). P (OD *vs*. ID) = 0.0873. **C** Experimental design of cell culture supernatant cytokine analysis: murine lung cells were extracted and plated in wells with or without plate bound anti-CD3 antibody and cultured for 24 h with or without soluble anti-CD28 antibody for T-cell activation. Cells were harvested and supernatant was collected, the latter was then analyzed using *LEGENDplex™ MU Th Cytokine Panel (12-plex)* kit. Samples were measured in *BD FACSymphony™ A1*. [*Created in BioRender. Harre, P. (2025)* https://BioRender.com/h86a453]. **D–I** Murine lung cell culture supernatant cytokine level measured with *LEGENDplex™ MU Th Cytokine Panel (12-plex)*. T-cell expansion was stimulated with plate bound anti-CD3-antibody and soluble anti-CD28-antibody in cell culture. **D** Supernatant IL-4 level in [pg/ml] (SD, standard diet n = 8; AD, arteriosclerosis diet n = 13; OD, ω-3 rich diet n = 8; ID, intervention diet n = 13). P (SD *vs*. OD) = 0.0377; P (AD *vs*. OD) = 0.0004; P (ID *vs*. OD) = 0.0122. **E** Supernatant IL-6 level in [pg/ml] (SD, standard diet n = 4; AD, arteriosclerosis diet n = 5; OD, ω-3 rich diet n = 4; ID, intervention diet n = 5). P (SD *vs*. OD) = 0.0286; P

(ID *vs*. SD) = 0.0042. **F** Supernatant IL-9 level in [pg/ml] (SD, standard diet n = 4; AD, arteriosclerosis diet n = 5; OD, ω-3 rich diet n = 4; ID, intervention diet n = 5). P (SD *vs*. OD) = 0.0375; P (ID *vs*. SD) = 0.0331. **G** Supernatant IL-10 level in [pg/ml] (SD, standard diet n = 4; AD, arteriosclerosis diet n = 5; OD, ω−3 rich diet n = 4; ID, intervention diet n = 5). P (AD *vs*. OD) = 0.0356. **H** Supernatant IL-17A level in [pg/ml] (SD, standard diet n = 4; AD, arteriosclerosis diet n = 8; OD, ω−3 rich diet n = 6; ID, intervention diet n = 8). P (AD *vs*. OD) = 0.0283. **I** Supernatant IL-5 level in [pg/ml] (SD, standard diet n = 4; AD, arteriosclerosis diet n = 5; OD, ω-3 rich diet n = 4; ID, intervention diet n = 5). P (SD *vs*. OD) = 0.0328; P (ID *vs*. SD) = 0.0358. **J, K** Flow cytometry was performed on blood and fresh total lung cells. **J** %CCR4+ cells gated on CD3+ CD4+ of total blood lymphocytes (SD, standard diet n = 4; AD, arteriosclerosis diet n = 5; OD, ω-3 rich diet n = 4; ID, intervention diet n = 5). P (SD *vs*. OD) = < 0.0001; P (ID *vs*. SD) = 0.0034. **K** %CCR4+ cells gated on CD3+ CD4+ of total lung lymphocytes (SD, standard diet n = 4; AD, arteriosclerosis diet n = 5; OD, ω-3 rich diet n = 4; ID, intervention diet n = 5). P (AD *vs*. OD) = 0.0103. **L** Correlation of %CCR4+ cells gated on CD3+ CD4+ of total blood lymphocytes with lung supernatant IL-5 levels (n = 18, P = 0.0006, R2 = 0.5303). Correlation **L** is shown by using simple linear regression. The two-tailed Pearson correlation analysis was performed to get the r and p value for **L**. Kruskal-Wallis test was used for (**B, G, H, K**). One-way ANOVA test was used for (**A, D–F, I, J**). All statistical tests were two-sided. No adjustments for multiple comparisons were applied. Data are shown as mean values ± s.e.m.; *P,0.05, **P,0.01, ***P,0.001, ****P < 0.0001.

cytometry revealed fewer CD3+ CD4+ CCR4+ cells in blood (Fig. 4J) and lung (Fig. 4K) of OD-fed mice (blood: 8.54%, lung: 1.66%, n = 4) versus SD (blood: 17.73%, lung: 2.05%, n = 4; P < 0.0001 and P = 0.0103, respectively). To confirm CD3+ CD4+ CCR4+ cell migration from lymphatic tissue to the blood and lung respectively, we therefore

correlated lung supernatant IL-5 levels and %CCR4+ gated on CD3+ CD4+ cells in the blood, observing a significant positive correlation (Fig. 4L).

All data considered, we concluded that administration of ω-3 rich PUFA diet and reduction of diet cholesterol led to decreased secretion of

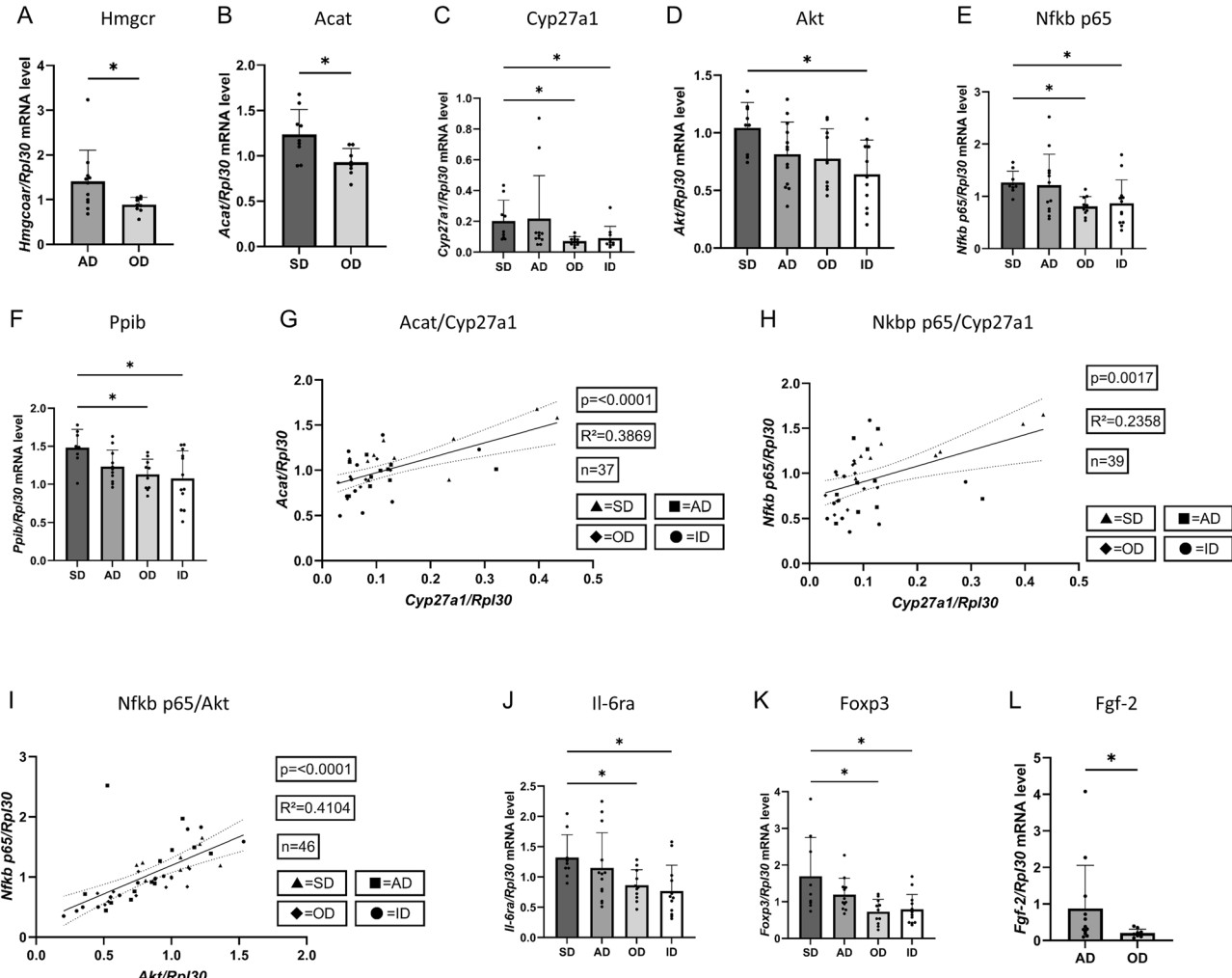

**Fig. 5 | Murine lung tissue gene expression level analysis of metabolic pathway enzymes connected to cholesterol via quantitative real-time PCR. A–F, J–L** RNA was isolated from the lung murine lung tissue. qPCR was performed and mRNA expression level was related to the housekeeping gene *Rpl30*. **A** *Hmgcr* mRNA expression in arteriosclerosis diet group (AD; n = 11) compared to ω-3 rich diet group (OD; n = 9). P (AD *vs*. OD) = 0.0310. **B** *Acat* mRNA expression in standard diet group (AD; n = 9) compared to ω-3 rich diet group (OD; n = 8). P (SD *vs*. OD) = 0.0129. **C** *Cyp27a1* mRNA expression in lung tissue of tumor bearing mice fed with standard diet (SD; n = 9), arteriosclerosis diet (AD; n = 11), ω-3 rich diet (OD; n = 10) and intervention diet (ID; n = 10). P (SD *vs*. OD) = 0.0235, P (SD *vs*. ID) = 0.0454. **D** *Akt* mRNA expression in lung tissue of tumor bearing mice fed with standard diet (SD; n = 9), arteriosclerosis diet (AD; n = 13), ω-3 rich diet (OD; n = 10) and intervention diet (ID; n = 11). P (SD *vs*. ID) = 0.0227. **E** *Nfkb p65* mRNA expression in lung tissue of tumor bearing mice fed with standard diet (SD; n = 9), arteriosclerosis diet (AD; n = 12), ω-3 rich diet (OD; n = 10) and intervention diet (ID; n = 12). P (SD *vs*. OD) = 0.0448, P (SD *vs*. ID) = 0.0485. **F** *Ppib* mRNA expression in lung tissue of tumor bearing mice fed with standard diet (SD; n = 8), arteriosclerosis diet (AD; n = 11), ω-3 rich diet (OD; n = 10) and intervention diet (ID; n = 12) P (SD *vs*. OD) = 0.0479, P (SD *vs*. ID) = 0.0295. **G** Correlation of relative *Acat/Rpl30* mRNA expression level with relative *Cyp27a1/Rpl30* mRNA expression level measured in murine lung tissue (n = 37, P = < 0.0001, $R^2$ = 0.3869).
**H** Correlation of relative *Cyp27a1/Rpl30* mRNA expression level with relative *Nfkb p65/Rpl30* mRNA expression level measured in murine lung tissue (n = 39, P = 0.0017, $R^2$ = 0.2358). **I** Correlation of relative *Akt/Rpl30* mRNA expression level with relative *Nfkb p65/Rpl30* mRNA expression level measured in murine lung tissue (n = 46, P = < 0.0001, $R^2$ = 0.4104). **J** *Il-6ra* mRNA expression in lung tissue of tumor bearing mice fed with standard diet (SD; n = 9), arteriosclerosis diet (AD; n = 13), ω-3 rich diet (OD; n = 11) and intervention diet (ID; n = 12). P (SD *vs*. OD) = 0.0423, P (SD *vs*. ID) = 0.0319. **K** *Foxp3* mRNA expression in lung tissue of tumor bearing mice fed with standard diet (SD; n = 9), arteriosclerosis diet (AD; n = 12), ω-3 rich diet (OD; n = 11) and intervention diet (ID; n = 13) in relation to the housekeeping gene *Rpl30*. P (SD *vs*. OD) = 0.0356, P (SD *vs*. ID) = 0.0443. **L** *Fgf-2* mRNA expression in arteriosclerosis diet group (AD; n = 12) compared to ω-3 rich diet group (OD; n = 9). P (AD *vs*. OD) = 0.0491. Correlations **G–I** are shown by using simple linear regression. The two-tailed Pearson correlation analysis was performed to get the r and *p* value for (**G–I**). Kruskal-Wallis test was used for (**C, D–F, K**). One-way ANOVA test was used for (**J**) and Mann-Whitney test was used for (**A, L**). Welch's t test was used for (**B**). All statistical tests were two-sided. No adjustments for multiple comparisons were applied. Data are shown as mean values ± s.e.m.; *P < 0.05.

cytokines associated with tumor growth, immune evasion and EMT, respectively metastatic tumoral features, within the TME.

## Dietary intake of ω-3 rich PUFAs counteracted accumulation of cholesterol and tumor driving cholesterol metabolite 27-HC in the TME

To confirm data obtained from our human cohort regarding gene expression of metabolic pathways of cholesterol, we measured gene expression levels of genes related to intracellular cholesterol accumulation and metabolism in lung tissue samples of mice fed with their corresponding diets. As described above, HMG-CoA-reductase (HMGCR), key enzyme in cholesterol neosynthesis and acetyl-CoA C-acetyltransferase 1 (ACAT1) which has been observed to increase intracellular cholesterol levels both have been reported upregulated in lung cancer cells by our department[2,41], hence we measured gene expression of the corresponding encoding gene via quantitative real-time PCR. As shown

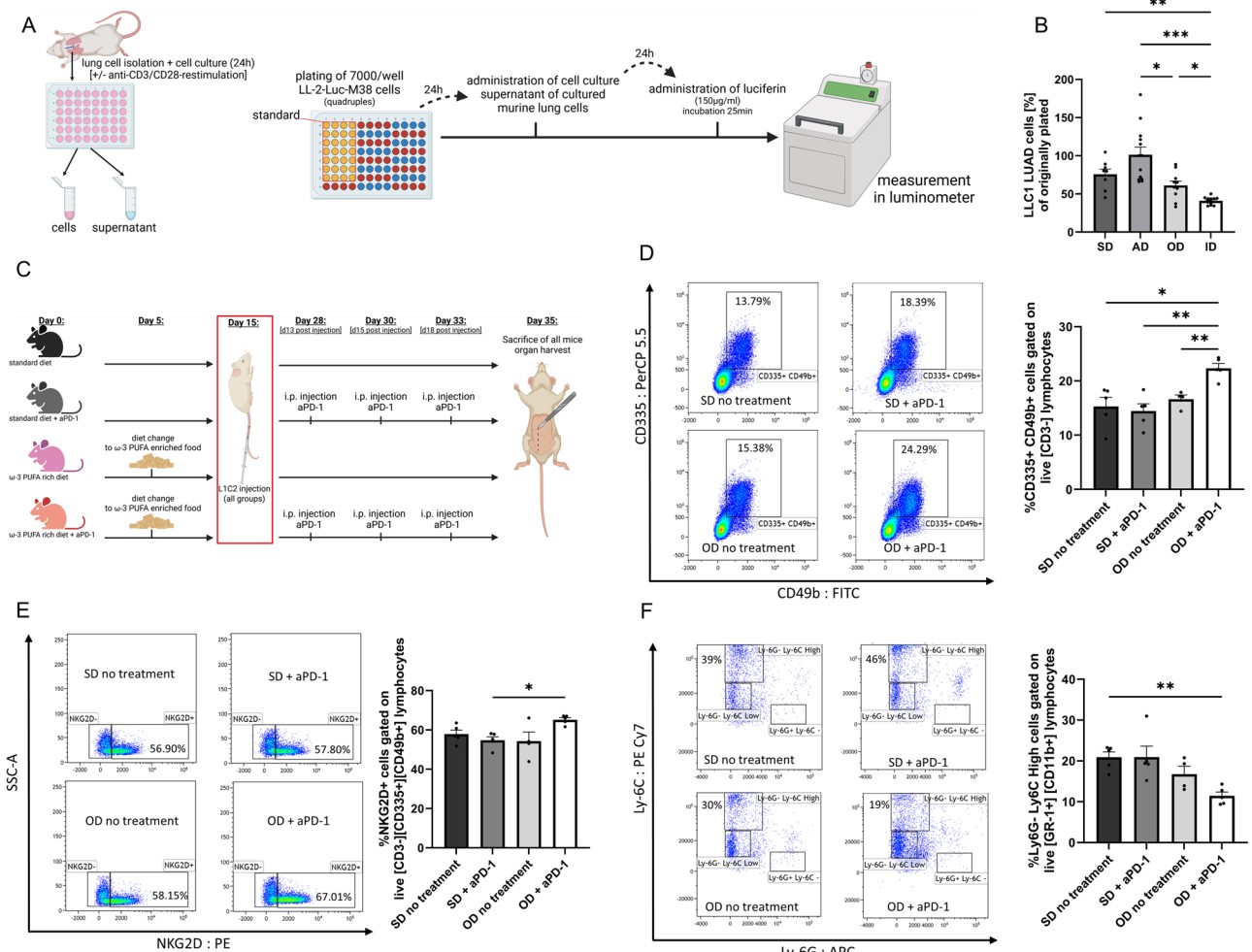

**Fig. 6 | Dietary ω-3 PUFAs enhance cytotoxicity of cytotoxic T lymphocytes (CTLs) and slow down tumor expansion in vitro. A** Experimental setup of bioluminescence assay. [Created in BioRender. Harre, P. (2025) https://BioRender.com/i70j158]. **B** Plotted bioluminescence assay data: the luminescence signal was converted to an estimated cell number via simple linear regression, data is given in % of originally plated cells (7000/well). Standard diet (SD; n = 8), arteriosclerosis diet (AD; n = 13), ω-3 rich diet (OD; n = 10) and intervention diet (ID; n = 13). P (SD *vs.* ID) = 0.0056; P (AD *vs.* OD) = 0.0139; P (AD *vs.* ID) = 0.0004; P (OD *vs.* ID) = 0.0345. **C** Experimental setup BALB/c murine NSCLC study. Mice were fed their indicated diet and injected tumor cells at 8 weeks of age. Mice within the therapy groups were administered aPD-1 intraperitoneal at day 13, 15 and 18 post tumor inoculation [Created in BioRender. Harre, P. (2025) https://BioRender.com/ghj67p9]. Flow cytometric analysis of lung (**D, E**) and spleen (**F**) tissue reveals enhanced anti-tumor immune responses in mice treated with aPD-1 and fed an ω−3 rich diet. **D–F** Flow cytometry was performed on fresh total spleen and fresh total lung cells. **D** Gating strategy and %CD335⁺ CD49b⁺ cells gated on live CD3- of total lung lymphocytes (SD, standard diet/no treatment n = 5; standard diet + aPD-

1treatment intra-peritoneal n = 5; OD, ω−3 rich diet diet/no treatment n = 4; OD, ω-3 rich diet + aPD-1treatment intra-peritoneal n = 5). P (SD no treatment *vs.* OD + aPD-1) = 0.0476; P (SD + aPD-1 *vs.* OD no treatment) = 0.0086; P (OD no treatment *vs.* OD + aPD-1) = 0.0085. **E** Gating strategy and %NKG2D⁺ cells gated on live CD3- CD335⁺ CD49b⁺ of total lung lymphocytes (SD, standard diet/no treatment n = 5; standard diet + aPD-1treatment intra-peritoneal n = 5; OD, ω−3 rich diet diet/no treatment n = 4; OD, ω−3 rich diet + aPD-1treatment intra-peritoneal n = 5). P (SD + aPD-1 *vs.* OD + aPD-1) = 0.0108. **F** Gating strategy and % Ly6G- Ly6C High cells gated on live GR-1⁺ CD11b⁺ of total splenic lymphocytes (SD, standard diet/no treatment n = 5; standard diet + aPD-1treatment intra-peritoneal n = 5; OD, ω−3 rich diet diet/no treatment n = 4; OD, ω−3 rich diet + aPD-1treatment intra-peritoneal n = 5). P (SD no treatment *vs.* OD + aPD-1) = 0.0033. Kruskal-Wallis test was used for (**B**). One-way ANOVA test was used for (**D–F**). All statistical tests were two-sided. No adjustments for multiple comparisons were applied. Data are shown as mean values ± s.e.m.; *P,0.05, **P,0.01, ***P,0.001.

in Fig. 5A, B, HMGCR and ACAT1 mRNA levels were significantly higher in AD-fed mice compared to OD. HMGCR was elevated in AD (n = 11) vs. OD (n = 9, P = 0.0310). ACAT1 was higher in SD (n = 9) vs. OD (n = 8, P = 0.0129). Moreover, we observed a general reduction of lung HMGCR mRNA expression in mice fed with ω-3 rich PUFA containing diet (OD, ID) when compared to standard diet (SD) and arteriosclerosis diet (AD) fed mice (Supplementary Fig. S5A) and noticed an overall reduction of ACAT1 gene expression levels in mice fed with ω-3 rich PUFA containing diets (OD, ID) (Supplementary Fig. S5B).

Taken together, we confirmed upregulated cholesterol neosynthesis and accumulation in the lung of mice that were administered standard diet (SD) or arteriosclerosis diet (AD).

Therefore, we next investigated whether or not accumulation of cholesterol within murine lung cancer cells is connected to hydroxylation of cholesterol via CYP27A1, generating 27-hydroxycholesterol (27-HC), thus activating downstream progressive, EMT-inducing and immune evasive pathways of lung cancer as described above. As displayed in Fig. 5C and Supplementary Fig. S5C respectively, CYP27A1 expression (Fig. 5C) was significantly lower in OD and ID compared to SD (P = 0.0235 and 0.0454, respectively) whereas overall gene expression levels of CYP27A1 showed to be generally reduced in mice fed with ω-3 rich PUFA containing diets (OD, ID). Consistent with our human data, we collected evidence of downstream pathways of 27-HC to confirm its immune suppressive and proliferative effect in the TME. Therefore, we first investigated AKT, NFkB p65 and PPIB

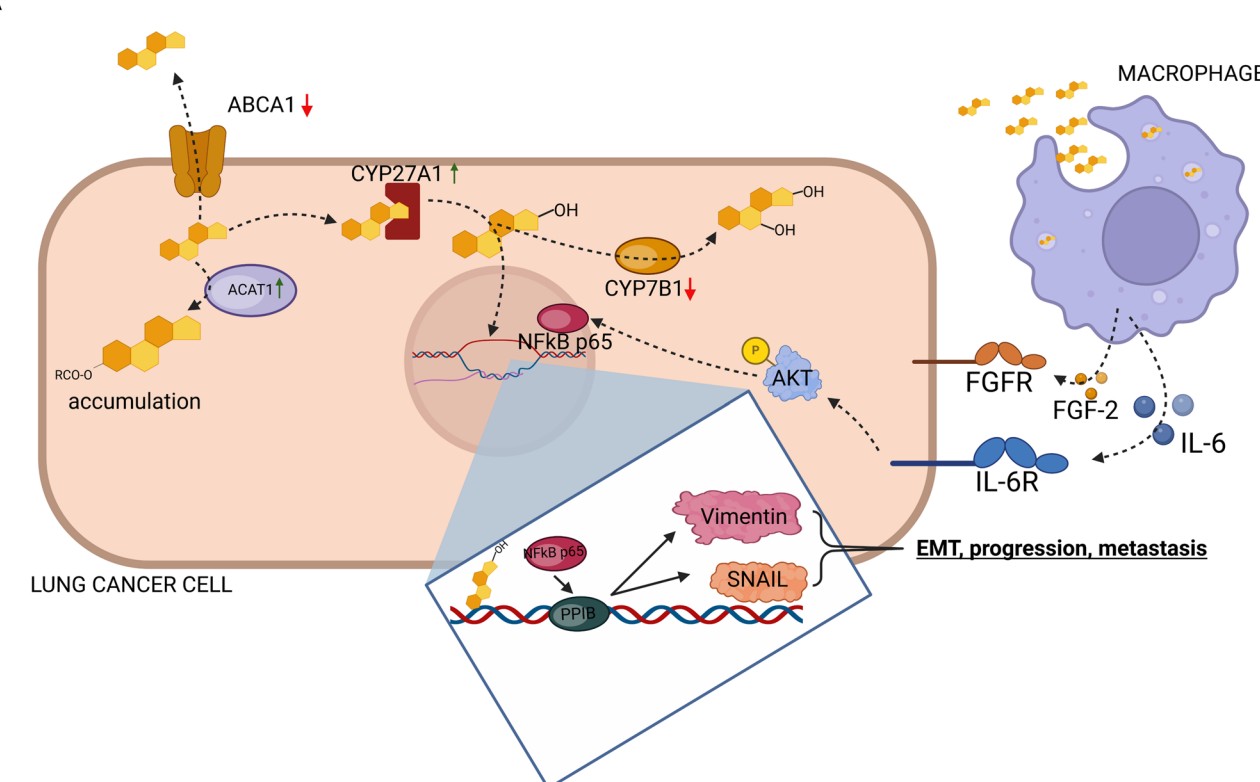

**Fig. 7 | Overview of cholesterol and its metabolite 27-hydroxycholesterol pathways in lung cancer cells.** [*Created in BioRender. Harre, P. (2025)* https://BioRender.com/s01e195]. We found ACAT1 and CYP27A1 upregulated and ABCA1 and CYP7B1 downregulated in both murine and human lung cancer cells, indicating cholesterol and cholesterol metabolite accumulation. CYP27A1 catalyzes hydroxylation of cholesterol to 27-hydroxycholesterol which induces NFkB p65 and PPIB, leading to EMT, progression and metastasis through induction of SNAIL and Vimentin. At the same time, secretion of IL-6 and FGF-2 by macrophages is promoted by this metabolite, stimulating the NFkB/PPIB axis through FGFR, IL-6R and AKT.

gene expression levels in lung tissue samples of mice fed with their corresponding diet. As stated above, these enzymes are strongly connected to proliferation, EMT and immune evasion and an induction through 27-HC has previously been reported[11,19–22,42–45]. Our analysis validated dietary influence on key 27-HC downstream effectors: AKT, NFkB p65, and PPIB were reduced in ω-3-fed groups. AKT (Fig. 5D) was lower in ID vs. SD (P = 0.0227); NFkB p65 (Fig. 5E) was lower in OD (P = 0.0448) and ID (P = 0.0485) vs. SD; and PPIB (Fig. 5F) was lower in OD (P = 0.0479) and ID (P = 0.0295) vs. SD. Furthermore, to evaluate intergenetic connections of cholesterol accumulation and metabolization, we correlated expression of ACAT and CYP27A1 in line with data acquired from our cohort. Here, consistently, gene expression correlations (Fig. 5G–I) confirmed these interactions: ACAT1 and CYP27A1 ($R^2$ = 0.387, P < 0.0001), CYP27A1 and NFkB p65 ($R^2$ = 0.236, P = 0.0017), and AKT and NFkB p65 ($R^2$ = 0.410, P < 0.0001).

In conclusion, these data confirmed the involvement of cholesterol metabolism and its derivatives in both human cohorts and a murine disease model.

### Dietary ω-3 rich PUFAs hindered induction of regulatory T cells and interfere with IL-6Ra expression while reducing FGF-2 in a murine model of LUAD

To validate tumoral mechanisms of immune evasion and chemoresistance via IL-6 signal transduction and induction of regulatory T cells, we proceeded to measure interleukin 6 receptor alpha subunit IL-6Ra in murine lung tissue samples via quantitative real-time PCR. IL-6Ra mRNA (Fig. 5J) was significantly reduced in OD (n = 11) and ID (n = 12) versus SD (n = 9; P = 0.0423 and 0.0319, respectively), suggesting impaired IL-6 mediated EMT, immune evasion and progression of NSCLC in mice fed with the corresponding diets. We next investigated forkhead box P3 (FOXP3)

expression levels in murine lung tissue of the corresponding diet groups. Not only is FOXP3 regarded as specific marker of immune suppressive regulatory T cells but as a direct promotor of metastasis and chemoresistance in NSCLC[46–48]. FOXP3 mRNA expression (Fig. 5K) was lower in OD (n = 11) and ID (n = 13) vs. SD (n = 9; P = 0.0356 and 0.0443). Next, we followed up on previously reported 27-hydroxycholesterol (27-HC) induced release of fibroblast growth factor 2 (FGF-2) by macrophages at tumoral site[11]. FGF-2 has been shown to activate proangiogenic features in cancerous cells while altering macrophage polarization, shifting them towards M2 immune suppressive phenotypes and has been declared a promising target of immune therapy concepts[49–51]. FGF-2 expression (Fig. 5L) was significantly reduced in the lung of OD-fed mice (n = 9) vs. AD (n = 12; P = 0.0491), implicating ω-3 PUFAs in inhibition of pro-angiogenic signaling, with mice fed OD generally having the lowest expression level of FGF-2 throughout all dietary groups (Supplementary Fig. S5E, F) leading to the assumption that dietary ω-3 PUFAs reduces macrophage secretion of FGF-2, hence impaired angiogenesis of cancerous cells and impaired macrophage polarization towards M2 phenotypes.

Taken together, here we demonstrated a ω-3 PUFAs mitigating effect on protumorigenic cascades mediated either by cytokines IL-6 and FGF-2 or regulatory T lymphocytes.

### Dietary ω-3 PUFAs enhanced cytotoxicity of cytotoxic T lymphocytes (CTLs)

To determine the influence of the dietary fatty acids on lymphocyte mediated cytotoxicity, we used an established model of cytotoxicity determination. Simulating a functional tumor microenvironment and investigating the functionality of the lung cell culture supernatant, we performed a bioluminescence cytotoxic assay ex vivo (Fig. 6A). Therefore, we challenged LUAD LL/2-luc-M38 cells with the supernatant of total lung cell cultures of

mice. As a reporter of surviving LL/2-luc-M38 cells, the luciferin mediated bioluminescence reaction was measured. Due to its conversion by luciferase expressed in living LL/2-luc-M38 cells, a highly induced luminescence signal represents higher numbers of living cells, and thus deceased cytotoxicity. Utilizing simple linear regression, the luminescence signal can then be converted to an estimated cell number. Here, we detected significantly decreased luminescence of LL/2-luc-M38 cells incubated with supernatant originating from αCD3/αCD28 antibody treated culture of lung cells obtained from mice fed with intervention diet (ID) (Fig. 6B).

We also noticed a significantly increased luminescence of LL/2-luc-M38 cells incubated with supernatant originating from αCD3/αCD28 antibody treated culture of lung cells from mice fed with arteriosclerosis diet (AD), when compared to all other diets that were used. Taken together, on the one hand, this analysis demonstrates a more induced anti-tumoral immune response in terms of induced cytokine secretion and cytotoxic T cell response through dietary ω-3 PUFA uptake (OD, ID), and an impaired immune and T cell response in mice administered cholesterol rich arteriosclerosis food (AD), on the other hand (Fig. 7).

### ω-3 PUFA-enriched diet enhances NK cell activity and reduces MDSC accumulation in the L1C2 lung cancer model

To strengthen the translational relevance of our findings, we incorporated an additional extensively used murine model, the BALB/c L1C2 lung cancer model[52,53], alongside the previously utilized human tissue samples and C57BL/6 LL/2 mouse model; in this model, we specifically investigated whether dietary administration of ω-3 PUFAs could enhance the anti-tumoral efficacy of PD-1 (programmed cell death protein 1) checkpoint inhibition (aPD-1).

Therefore, BALB/c mice were assigned to one of four experimental groups based on a two-factor design: diet type (standard diet or ω-3 PUFA-enriched diet, Supplementary Tables S5, S6) and treatment condition (with or without aPD-1 administration). This resulted in four distinct groups, as depicted in Fig. 6C. The diets did not differ from those used in previous experiment with the C57BL/6 LL/2 mouse model. Ten days following the initiation of dietary intervention, all groups of mice were intravenously injected with L1C2 cells. Mice assigned to the treatment condition were subsequently administered the checkpoint inhibitor aPD-1 intraperitoneally on days 13, 15, and 18 post-tumor cell injection. On day 35, after ending the experiment, the lungs and spleens of the mice were removed, and cells were isolated and analyzed via flow cytometry.

Here, we focused on natural killer (NK) cells and myeloid-derived suppresser cells (MDSCs) and their presence in the murine lung and spleen depending on the diet and treatment the mice received.

Natural killer (NK) cells are defined by the absence of CD3 expression and the presence of CD335 and CD49b surface markers[54–56]. Accordingly, we quantified CD335+ CD49b+ lymphocytes in relation to live CD3- lymphocytes in lung tissue and observed a significant increase in NK cell frequency in the group receiving both the ω-3 PUFA-enriched diet and aPD-1 treatment compared to all other experimental groups (Fig. 6D). To further assess NK cell activity, we analyzed the expression of the activating receptor NKG2D on live CD3- CD335+ CD49b+ lymphocytes. NK cells from the group receiving the ω-3 PUFA-enriched diet combined with aPD-1 treatment exhibited the highest NKG2D expression among all experimental groups, with a statistically significant increase compared to the standard diet group treated with aPD-1 (Fig. 6E).

Myeloid-derived suppressor cells (MDSCs) are a heterogeneous population of immunosuppressive myeloid cells known to inhibit T cell responses and promote tumor progression, and their accumulation has been associated with poorer clinical outcomes and reduced survival in cancer patients[57–59]. Among them, monocytic MDSCs are commonly characterized by a Ly6G- Ly6C$^{high}$ phenotype within the Gr1+ CD11b+ compartment. In our analysis, we evaluated this population in the spleen in relation to treatment effects. We found that the frequency of live Ly6G- Ly6C$^{high}$ Gr1+ CD11b+ MDSCs was significantly reduced in mice receiving the ω-3 PUFA-enriched diet combined with anti-PD-1 therapy, compared to the untreated standard diet group (Fig. 6F). In conclusion, the ω-3 PUFA–enriched diet combined with aPD-1 treatment was associated with a concurrent increase in NK cell frequency and activity, along with a significant reduction in splenic MDSC levels. These changes indicate a coordinated shift toward increased cytotoxic potential and decreased immunosuppressive cell populations in this model.

## Discussion

Diet has become one of the biggest healthcare related challenges in the modern western society. Many people suffer from diseases which can directly be connected to nutrition components and eating habits, such as disproportionately high sugar and fat intake. Shedding light on the contribution of dietary cholesterol and ω-3 PUFAs' to the development of non-small cell lung cancer, one of the leading causes of cancer-related death worldwide, and how dietary interventions could be utilized to support modern immune therapies in cancer is highly relevant. Therefore, improving patients' overall survival rate by influencing tumor growth, metastasis and chemoresistance on a molecular level at the same time as stabilizing quality of life aspects regarding side affects of the applied therapies was the goal of the present study. Hence, we analyzed the role of cholesterol and ω-3 PUFAs in patients with lung adenocarcinoma (LUAD), an experimental murine model of LUAD and in vitro human cell culture experiments.

Cholesterol is an essential component of mammalian cell membranes and plays an utterly important role in inter- and intracellular signal transductions. However, it has become more and more apparent that abnormal raised amounts of cholesterol are positively correlated with a higher risk of developing various types of cancer[60–62]. Thus, targeting cholesterol metabolism has emerged as supporting therapy concept when preventing and treating cancer[63]. Nevertheless, the role of cholesterol in lung cancer is still insufficiently understood and controversely discussed. While some studies claim low plasma cholesterol concentrations to be associated with increased lung cancer risk and poor clinical outcome, others observed a close connection of raised cholesterol levels and lung cancer incidence[2,64–66]. Recently, metabolites of cholesterol, most prominent 27-hydroxycholesterol (27-HC) have attracted attention in the field of cancer research[67]. 27-HC has linked high fat malnutrition to various types of cancer, predominantly breast cancer as its structure enables direct estrogen receptor stimulation in the latter[21,68]. In lung cancer however, Li et al. were able to disclose relevance of cholesterol, respectively 27-HC, regarding tumor growth and metastasis in a murine CYP27A1 knockout model and in vitro experiments[11]. Here, they demonstrated inhibited lung tumor growth in CYP27A1 knockdown mice on the one hand and 27-HC mediated phosphorylation of AKT and NFkB p65, PPIB production and increased FGF-2 and IL-6 secretion, thus tumor progression and metastasis on the other hand. To verify these data, we analyzed a cohort of patients with histologically confirmed lung adenocarcinoma. Previously, our group described altered cholesterol metabolism that led to intracellular cholesterol accumulation and a cholesterol rich TME in control and lung tumor tissue acquired from patients with LUAD[2]. In this study, we could further reveal downstream metabolization of accumulated cholesterol. We were able to directly correlate earlier reported ACAT1 with CYP27A1 mRNA expression levels, indicating influx dependent cholesterol hydroxylation. In line with Li et al., we observed increased CYP27A1 and decreased CYP7B1 mRNA levels in the tumoral area of our human cohort of patients with diagnosed non-small cell lung cancer when compared to the corresponding control area[11]. Furthermore, we correlated CYP7B1 with the resected tumor diameter and found an inverse correlation. Hence, we concluded that tumor cells accumulate cholesterol through increased influx (ACAT1) and reduced efflux (ABCA1). The accumulated cholesterol is then converted into 27-HC, which also accumulates due to the downregulation of CYP7B1. Whether or not CYP27A1 is induced or suppressed by dietary lipids could not be demonstrated in our human cohort as we lacked data of patients' dietary food composition.

In tumor, there are many molecular mechanisms that have been uncovered in the past regarding epithelial mesenchymal transition and immune evasion, both of which are regarded as main challenges in modern

cancer therapy. Sanaei et al. reviewed the PI3K/AKT/mTOR signaling pathway and considered alterations in this axis to consequently lead to oncogenic features in cells of the lung[43]. Moreover, Bu et al. most recently reported high-fat diet to promote AKT mediated liver cancer[69] and several studies (Li et al.[11], Vurusaner et al.[70]) reported **27**-HC related phosphorylation of AKT, resulting in activation of downstream signaling and induction of NFkB. NSCLC has been shown to upregulate PD-L1 expression in order to suppress immune responses, thus immune therapies targeting these immune checkpoints have emerged[71]. However, only a small percentage of patients respond to this kind of therapy. NFkB is highly suspected to mediate PD-L1 transcription in lung cancer[44]. There is evidence that 27-HC is directly involved in induction of NFkB, hence downstream activation of immune suppressive pathways[11,72]. Cyclophilin B, encoded by PPIB, is linked to cellular transcriptional regulation, proliferation and immune response[42]. Overexpression of NFkB p65 is highly connected to immune evasion of NSCLC and its progression while the PI3K/Akt/mTOR pathway is probably the most altered pathway in NSCLC, resulting in epithelial-mesenchymal transition (EMT), proliferation and drug resistency[43–45]. Hwang et al. were able to demonstrate that cyclophilin B plays a vital role in cell proliferation, colony formation and migration of lung adenocarcinoma cells[42]. Signal transduction via AKT/NFkB is suspected to be a primary driver of NSCLC progression and has already been recognized as potential therapeutic target[29]. In our study, we therefore measured AKT, NFkB and PPIB mRNA expression levels in tumoral and control tissue samples of patients with histologically confirmed lung adenocarcinoma and found expression levels not only upregulated in tumor but significantly and directly correlated to CYP27A1 mRNA expression levels, thus confirming a close connection of cholesterol metabolites, tumoral progress and immune suppression.

Taken together, analysis of our human cohort gave insight on intracellular metabolic pathways of cholesterol in human lung cancer cells. Thus, we established a murine model of cancer in which observed differences could be directly linked to diet. In the past decades, ω-3 PUFAs have been advocated to reduce cancer risk in various cancer types and some studies even suggest antiproliferative effects of ω-3 PUFAs on lung tumor growth[17,73–75]. Moreover, in vitro experiments with human lung adenocarcinoma cell lines found downregulation of adhesion, invasion and growth-related genes through administration of ω-3 PUFAs[76]. However, exact molecular mechanisms remain veiled and studies on therapeutic effects of ω-3 PUFAs in the treatment of NSCLC are scarce. To contrast high fat high cholesterol diet (arteriosclerosis diet, AD), we therefore decided to introduce a Mediterranean type of diet (ω-3 PUFA rich diet, OD) and a mixed dietary concept (intervention diet, ID) to our model of murine lung cancer to further evaluate advantages and disadvantages of both diets. The expected shift towards a typical western serum lipid profile (high serum LDL, low serum HDL) in the arteriosclerosis diet group on the one hand and lower serum LDL but high HDL in mice fed ω-3 PUFA containing food on the other hand was confirmed by serum lipid analysis. In vivo imaging and tumor load evaluation of histologic slides taken from the lung showed increased tumor load in mice that were fed with a high fat diet. Simultaneously, we observed that mice fed with dietary ω-3 PUFAs developed significantly less tumor than mice fed with either the standard diet (SD) or arteriosclerosis diet (AD). In addition, we noticed enhanced cytotoxicity and anti-tumoral activity in ω-3 PUFA administered mice which is reflected by bioluminescence cytotoxicity assays we performed, increased serum IFN-γ levels and reduced IL-10 secretion in total lung cell supernatant. We previously identified IL-10 as counteractor on the effect of IFN-γ on the PD1/PDL1 pathway, thus resulting in tumoral resistance to anti-PD1/PDL1 immunotherapy[77,78]. Wu et al. demonstrated that subtypes of ω-3 PUFAs might be linked directly to upregulation of IFN-γ release and enhanced cytotoxic activity of NK cells in a murine model of melanoma[79]. The reason why IFN-γ secretion in our study was only induced in mice fed consistently with ω-3 PUFA rich food (OD group) remains unclear. In addition, we found increased serum levels of TNFα in mice fed with arteriosclerosis and ω-3 rich PUFA diet, whereas mice that were administered interventional

diet (ID) showed the lowest concentration of TNFα. This might emphasize the dual role of cholesterol in lung cancer being essential for clonal expansion of T-cells and other vital mechanisms of immune transduction on the one hand, but on the other hand also driving cancer cell immune evasion and tumor expansion[64,80–82]. Moreover, ω-3 PUFA administration seems to have various effects on IL-10 release, depending on the pathological context. In inflammation, there is evidence that ω-3 PUFA induces IL-10 production, thus exerting immune suppressive effects[83]. Similar effects were observed in a model of murine melanoma but not NSCLC[84]. As stated above, we consider IL-10 an immunosuppressive cytokine playing a key role in tumor immune escape. Accordingly, in the present study we demonstrated reduced IL-10 levels in the supernatant of total lung cells of mice that were administered ω-3 PUFAs as immune sensitization mechanism. Mice that were administered a high fat high cholesterol diet suffered from highest tumor load and showed least cytotoxic activity. Extensive research has displayed the negative effect of cholesterol on intrinsic anti-tumoral cytotoxicity and recently it has been revealed that cholesterol enriched membranes in cancer cells impair immune cell mediated cytotoxicity at a biomechanical level[7]. Consistent with existing literature, we therefore concluded that high fat high cholesterol diet acts immune suppressive and impairs cytotoxicity in the TME of lung cancer.

Next, we investigated whether or not dietary lipid composition influences cholesterol and oxysterol metabolism in a model of murine lung adenocarcinoma. In line with our human cohort data, we analyzed mRNA expression levels of HMGCR, a key enzyme in cholesterol neosynthesis, and ACAT1 and found both to be reduced in mice that were consistently fed with ω-3 PUFA rich diet (OD), leading us to the assumption that the latter might impair accumulation of cholesterol in lung cancer cells. Moreover, we observed that mice fed with ω-3 PUFA rich diet showed the lowest CYP27A1 mRNA expression levels which correlated with ACAT1 mRNA expression levels. This emphasizes a close connection between dietary lipid composition, intracellular cholesterol accumulation and hydroxylation. As discussed above, we suspect that cholesterol hydroxylation might, among other factors, be causal for phosphorylation of AKT and the activation of NFkB p65 and PPIB. Simultaneously, administration of ω-3 PUFAs has been shown to reduce AKT phosphorylation levels in models of breast and colorectal cancer[85,86] and was reported to antagonize NFkB signaling in various types of inflammatory diseases[87,88]. In the present study, we found AKT, NFkB p65 and PPIB mRNA expression levels to be significantly lower in mice fed with ω-3 PUFA rich diet, thus negatively affecting oncogenic alteration in these signaling pathways. Regarding the influence of cholesterol on this axis we remain unable to make a statement here as there was no significant difference in the expression levels between the standard diet and cholesterol rich diet.

We next sought to evaluate the influence of dietary fatty acids on other cancer favoring cytokine release profiles in lung cancer. In this context, IL-4 is suspected to enhance tumor-associated inflammation and promote angiogenesis while skewing the immune response away from anti-tumoral Th1 response towards immunosuppressive Th2 response. Evidence on the influence of dietary lipids on IL-4 release in cancer is limited and spreads across various types of neoplastic diseases. However, these findings need to be interpreted with caution in the context of lung cancer, as the TME can vary significantly between different cancer types. In the present study, we noticed no visible influence of cholesterol on the secretion of IL-4 in the supernatant of total lung cells. By contrast, we observed a significant reduction of IL-4 release in mice fed with ω-3 PUFA rich diet. Recent studies suggest a pro-tumorigenic effect of IL-4 by signal transduction via the IL-4/STAT6 axis resulting in polarization of tumor associated macrophages (TAM) and increasing numbers of cancer driving M2 myeloid cell phenotype[89,90]. Consequently, we inferred that in NSCLC, administration of ω-3 PUFAs might counteract on IL-4 mediated Th2 shifting of immune cells within the TME.

Data confirming IL-6 as contributor to EMT and metastasis in lung and other types of cancer is abundant. A recent meta-analysis by Guo et al. demonstrated that ω-3 PUFA supplementation significantly reduced IL-6

levels in patients with diagnosed cancer[31–35,91]. Whether or not dietary cholesterol intake has any influence on IL-6 release in patients with cancer is largely unknown. In the present study, we observed reduced release of IL-6 by total lung cells of mice that were administered ω-3 PUFAs in their diet, implying mitigation of IL-6 induced EMT and metastasis in these mice. Measuring IL-6 receptor expression in lung tissue of these mice substantiated our proposition.

Currently, there is insufficient direct evidence to suggest that ω-3 PUFA administration alters IL-9 secretion generally or in the context of lung cancer. However, high cholesterol levels might favor differentiation of T helper cells into Th2 subsets which share signaling pathways with Th9 cells, thus linking IL-9 release and high cholesterol levels. In the present study, we did not observe any significant differences of IL-9 release when comparing standard and arteriosclerosis diet. Nevertheless, we were able to demonstrate a significant reduction of IL-9 release in mice fed with ω-3 PUFAs.

The function of IL-17A in lung cancer is a topic of debate in academic publications. As previously mentioned, IL-17A on the one hand promotes tumor growth and angiogenesis in lung cancer but is also vital for neutrophile and macrophage recruitment into the TME on the other hand. There is no direct evidence that IL-17A secretion is influenced by either cholesterol or ω-3 PUFAs with regard to lung cancer. However, recent studies suggest that administration of ω-3 PUFAs inhibits IL-17A secretion in adipose tissue[92]. High cholesterol levels are associated with systemic inflammation, thus stimulation of IL-17A release[93]. Previously, it was aimed to shed light on this potential angiogenic factor in NSCLC and it was reported that in vivo targeting of IL-17A resulted in abrogated tumor growth and inhibition of CD4+ CD25+ FOXP3+ T regulatory cells while inducing IFN-γ and TNFα production in CD4+ cells[53]. This might be another possible link between high fat high cholesterol diet and cancer progression. While examining IL-17A secretion levels in the corresponding diet groups utilized in this study, we found them induced in the arteriosclerosis diet group when compared to other diets, implicating systemic inflammation, consequently IL-17A release caused by high cholesterol levels.

Effects of dietary ω-3 PUFA on IL-5 secretion in the lung are discussed controversially in the literature and evidence in the context of cancer is rather rare. In children with asthma, Farjadian et al. demonstrated decreased IL-5 production by ω-3 PUFA substitution[94]. In contrast, Yin et al. claimed enhanced IL-5 secretion via dietary administration of ω-3 PUFAs in a murine model of allergic lung inflammation[95]. Dietary cholesterol influence on IL-5 production in the lung seems to be equally concealed as studies observed a connection of high dietary cholesterol intake and IL-5 mediated inflammation in asthmatic murine lungs[96,97].

In our model of cancer, we found IL-5 release not only to be directly connected to dietary lipid composition but decreased in mice with high ω-3 PUFA intake. In lung cancer, IL-5 induction might be involved in trafficking of immune suppressive regulatory T cells into the TME via CCL22 release, acting as chemoattractant for CCR4+ T cells[40]. At the same time, we found FOXP3 mRNA expression to be downregulated in mice fed with ω-3 PUFA containing diet, emphasizing the influence of ω-3 PUFAs on the TME. We thus concluded that dietary ω-3 PUFA avert IL-5 mediated immune suppression and flow cytometric determination of the proportion of CD3+ CD4+ CCR4+ Tregs in the blood and lung undermined our assumption.

In addition to analyzing metabolic enzyme pathways, cytokine profiles, and T cell responses, we extended our investigation of ω-3 PUFAs to other key components of the immune system, specifically natural killer (NK) cells and myeloid-derived suppressor cells (MDSCs). To strengthen and expand our findings, we next used the BALB/c L1C2 lung cancer model, which provides a genetically and immunologically distinct background from the C57BL/6 LL-2 system. This model allowed us to specifically investigate the effects of dietary ω-3 PUFAs, alone and in combination with PD-1 checkpoint inhibition, on NK cell activation and MDSC accumulation. In the L1C2 model, dietary administration of ω-3 PUFAs significantly enhanced NK cell frequency and activation in the lung, as evidenced by increased expression of the activating receptor NKG2D, particularly when combined with anti-PD-1 (aPD-1) checkpoint therapy. Simultaneously, this combination therapy led to a marked reduction in splenic monocytic MDSCs (Ly6G- Ly6Chigh Gr1+ CD11b+), indicating reduced immune suppression due to lower MDSC levels, accompanied by potentially increased antitumor activity through enhanced NK cell function in the tumor-bearing host. These findings are consistent with previous reports indicating that ω-3 PUFAs can modulate NK cell function, enhancing cytotoxic activity while mitigating immunosuppressive influences in the tumor microenvironment. Tumor-associated NK cells in NSCLC are often functionally impaired. Platonova et al. demonstrated that intratumoral NK cells exhibit reduced degranulation and diminished IFN-γ production due to downregulation of activating receptors such as NKp30, NKG2D, and DNAM-1[98]. Similarly, Krneta et al. showed that NK cells infiltrating breast tumors display an immature phenotype marked by reduced perforin, granzyme B, and NKp46 expression—features that could be reversed by IL-12 and anti-TGF-β treatment[56]. Our data suggest that ω-3 PUFAs may counteract this suppressive tumor environment by supporting NK cell maturation and activation, potentially via modulation of cytokine signaling pathways such as IL-6. High levels of monocytic MDSCs (M-MDSCs) have been consistently linked to poor prognosis and resistance to immunotherapy in NSCLC patients[99]. Koh et al. demonstrated that M-MDSCs impair the efficacy of anti-PD-1 therapy through secretion of IL-6, IL-10, and TGF-β, leading to T-cell exhaustion and suppressed cytotoxic responses[58]. Jeong et al. recently further highlighted IL-6 as a central mediator of MDSC recruitment and activation in PD-L1-high tumors, promoting resistance via the IL-6/Jak2/Stat3 axis[59]. Given that ω-3 PUFA administration in our study was associated with reduced IL-6 secretion, increased NK cell activation, and diminished MDSC accumulation, we propose that the immunomodulatory effects of ω-3 PUFAs are at least partially mediated through suppression of IL-6-dependent MDSC signaling. Together, these results highlight a promising combinatorial strategy in which dietary ω-3 PUFAs enhance innate immune function and improve the efficacy of checkpoint inhibition by alleviating tumor-induced immunosuppression in NSCLC

In spite of these new discoveries, the limitations and potential improvements of the methods used must be discussed. Patients acquired for this study were therapy naive and their nutritional status is unknown. Given the limitations within our human cohort, we were unable to further investigate the link between dietary lipid composition and cholesterol metabolism. Consequently, we turned to an animal model to address this question. Data obtained from animal models do not fully replicate either the complexity of human lung cancer and its pathological varieties, nor are the conclusions drawn from this data always applicable to clinical practice. Moreover, we discussed many metastatic pathways in relation to the murine model of LL-2 induced lung cancer that we have used, however LL-2 is a cell line that is considered only moderately metastatic. Predictions about metastatic potential might thus not be valid, further studies should aim to validate these findings in different models of lung cancer.

Taken together, we concluded that dietary lipid composition acts on cytokine release in lung cancer TME. Moreover, we were able to demonstrate that dietary ω-3 PUFAs regulate cytokine release within the TME, potentially shifting secretion towards a less immune evasive, less metastatic pattern. In addition, ω-3 PUFAs enhanced the efficacy of aPD-1 checkpoint therapy, which was associated with increased NK cell activation and reduced accumulation of immunosuppressive MDSCs.

On the contrary, dietary cholesterol and its metabolites accumulate in cancer cells and impair immune response and reduce cytotoxicity of immune cells in the TME. Although further clinical validation is necessary, these findings highlight the potential of dietary interventions as a therapeutic adjunct in NSCLC treatment.

Future studies should investigate the specific mechanisms by which ω-3 PUFAs and cholesterol modulate immune cell populations and released cytokines in the NSCLC microenvironment.

## Data availability

The numerical data plotted (source data) in Figs. 1–6 as well as S3 and S5 are provided in Supplementary Data 1. This study includes

patient-derived clinical and molecular data, murine tumor model data, and in vitro assay data. Patient data contain sensitive clinical information and are therefore not publicly available; access may be granted for academic purposes upon reasonable request to the corresponding author, subject to institutional and ethical approvals. All mouse experiment datasets, flow cytometry files (FCS), cytokine ELISA results, qPCR data, and imaging datasets supporting the findings of this study are available from the corresponding author upon request. All data are stored securely on institutional servers at Friedrich-Alexander-Universität Erlangen–Nürnberg, with regular backup.

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

## Acknowledgements

This work was supported by SFB1181 in Erlangen and by the Department of Molecular Pneumology, and P.H.'s fellowship was financially supported by the CRC1181 in Erlangen. Moreover this work was supported by a DFG grant FI817/6 awarded to S. F. We thank the team of the Department of Molecular Pneumology; Sonja Trump, Adriana Geiger, Susanne Mittler and Elvedina Nendel for their excellent technical help. We also thank the Department of Pathology team and the team at the Thoracic Surgery, as well as the Department of Medicine 1 of the University Hospital Erlangen. The present work was performed in fulfilment of the requirements for obtaining the degree "Dr. med." for Philipp Harre.

## Author contributions

Conception and design: P. Harre and S. Finotto. Acquisition of data: P. Harre, D. I. Trufa, K. Hildner, M. T. Chiriac, C. I. Geppert, and H. Sirbu. Analysis and interpretation of data: P. Harre, K. Hohenberger, P. Tausche, and S. Finotto. Writing and review: P. Harre and S. Finotto. Technical and material support: P. Harre, D. I. Trufa, M. T. Chiriac, C. I. Geppert, K. Hohenberger, S. Krammer, Z. Yang, P. Tausche, H. Sirbu, and S. Finotto. Study supervision: S. Finotto.

## Funding

## Competing interests

The authors declare no competing interests.

## Additional information

[1]Department of Molecular Pneumology, Friedrich-Alexander-Universität (FAU) Erlangen-Nürnberg, Universitätsklinikum Erlangen, Erlangen, Germany. [2]Department of Thoracic Surgery, Friedrich-Alexander-Universität (FAU) Erlangen-Nürnberg, Universitätsklinikum Erlangen, Erlangen, Germany. [3]Bavarian Cancer Research Center (BZKF), Erlangen, Germany. [4]Comprehensive Cancer Center Erlangen-EMN (CCC ER-EMN), Erlangen, Germany. [5]Department of Medicine 1—Gastroenterology, Pneumology and Endocrinology, Friedrich-Alexander-Universität (FAU) Erlangen-Nürnberg, Universitätsklinikum Erlangen, Erlangen, Germany. [6]Institute of Pathology, Friedrich-Alexander-Universität (FAU) Erlangen-Nürnberg, Universitätsklinikum Erlangen, Erlangen, Germany. [7]Department of Paediatrics and Adolescent Medicine, Friedrich-Alexander-Universität (FAU) Erlangen-Nürnberg, Universitätsklinikum Erlangen, Erlangen, Germany. [8]Deutsches Zentrum für Immuntherapie (DZI), Erlangen, Germany. ✉e-mail: Susetta.Finotto@uk-erlangen.de

