## [Transparent Peer Review file · Communications Medicine]

Dietary ω -3 polyunsaturated fatty acids (PUFAs) reduce cholesterol-driven non-small cell lung cancer (NSCLC) progression in mouse models.

Corresponding Author: Professor Susetta Finotto

Version 0:

Reviewer comments:

Reviewer #1

(Remarks to the Author)

The manuscript investigates the interplay between dietary lipids, specifically high cholesterol and omega 3 polyunsaturated fatty acids (PUFAs), and lung cancer progression. Using a combination of human NSCLC tissue analyses and a murine lung cancer model, the authors provide evidence that a high cholesterol diet is associated with increased tumor load, immune evasion (via altered cytokine profiles and regulatory T cell recruitment), and activation of oncogenic pathways. Conversely, omega 3-PUFA supplementation appears to reduce these protumorigenic signals and enhance cytotoxic responses. Overall, the work is relevant to both oncology and nutritional science and may open new avenues for adjunct dietary interventions in NSCLC management.

Major comments

1. The study relies solely on intravenous injection of LL/2 Luc M38 cells, which may limit the generalizability of the findings. Testing an additional cell line model, such as 393P lung adenocarcinoma cells derived from KrasLA1/+; p53R172HΔG mice, would help validate and strengthen the conclusions.
2. I was surprised that the manuscript did not evaluate the potential impact of omega 3 PUFAs on the efficacy of immune checkpoint inhibitors. Given the prominence of anti PD 1 and CTLA 4 therapies in NSCLC, exploring whether omega 3 PUFAs could synergize with or potentiate these treatments would substantially enhance the study's translational relevance.
3. Both the arteriosclerosis diet (AD) and the ω 3 PUFA diet (OD) contain approximately 20% fat; however, the manuscript does not specify the cholesterol content in AD. Without a high fat, cholesterol free control diet, it is difficult to isolate the specific effects of cholesterol from those of fat, saturated fats, or overall caloric density. AD is described as "cholesterol rich" but lacks absolute values (e.g., mg/kg or % w/w cholesterol), and it is unclear whether OD includes any cholesterol. Clarifying these points and including a control group with a high fat, cholesterol free diet would allow for a more precise interpretation of the results.
4. Although the human sample analysis reveals altered cholesterol metabolism in NSCLC tissues, the absence of detailed dietary intake information, such as food frequency questionnaires or 24 hour dietary recalls, and comprehensive nutritional biomarkers hinders the establishment of a direct link between these molecular changes and patients' dietary habits. Without this information, it remains unclear whether the observed alterations in cholesterol related gene expression and metabolism are driven by dietary lipid intake or by other factors (e.g., underlying disease processes, metabolic conditions, or medication use). Including retrospective or prospective dietary data collection from the human cohort would help correlate nutrient intake with tumor gene expression profiles and clinical outcomes.
5. While the manuscript demonstrates that dietary omega 3 PUFAs are associated with enhanced CTL cytotoxicity in vitro and reduced induction of regulatory T cells, with concurrent decreases in IL 6Ra and FGF 2 levels in a murine LUAD model, the findings remain largely correlative. The data suggest an association between omega 3 PUFA intake and improved anti tumoral immune responses but do not fully elucidate the underlying cellular and molecular mechanisms in the tumor immune microenvironment. To strengthen the mechanistic insight, additional experiments are required. Comprehensive immunophenotyping of the tumor microenvironment (e.g., using flow cytometry to quantify and characterize tumor infiltrating

lymphocyte subsets such as CD8+ T cells, regulatory T cells, NK cells, and myeloid derived suppressor cells) would provide more direct evidence of how dietary omega 3 PUFAs reshape the immune landscape and contribute to anti tumoral activity.

Minor comments

1. Correct typographical errors in the title (e.g., “faciliated” should be “facilitated”).
2. Ensure consistent use of abbreviations throughout the manuscript, standardizing between “ ω 3 PUFAs” and “omega 3 PUFAs.”

Amr Khalifa
AIRC-Foundation for Cancer Research Post-Doc Fellow
University of Genoa, Italy

Reviewer #2

(Remarks to the Author)

The study led by Harre et al. aimed to assess the role of dietary lipids, cholesterol, and omega-3 polyunsaturated fatty acids within lung cancer and their potential role in modulating cancer treatment. Below are my comments and suggestions.

Comments for the Author:

Major Concerns:

1. Study experimental system: This study had two components: an analysis of lung tissue samples from 152 patients and a mice model study, yet there is no indication of these experimental systems in the study title, abstract, or introduction. It is not until line 180 that in vivo is mentioned and the word mice is written for the first time on line 179. The manuscript mentions that the study will look at cohorts of patients at the end of the introduction (line 113), yet it is not clear what the multiple cohorts were. The methods section stated that study data were provided by the University of Erlangen’s hospital, yet the methods do not mention multiple cohorts. The experimental systems used to study the disease mechanisms should be clearly explained and described in the initial manuscript pages.
2. Introduction: The manuscript discusses omega 3-PUFAs but does not provide any details on what these are. As this is a cornerstone of the manuscript, including details on omega PUFAs would be important. Similarly, while the lung tumor microenvironment is included as part of the research objective, it is unclear what a TME actual is. A clear explanation of the concepts central the to study would improve the manuscript overall.
3. Results:
 - a. This section should begin with a description of the Table 1 results, for readers to understand the study population and the distribution of its characteristics. While Table 1 is referenced on line 132 and included in the manuscript (line 808), the manuscript does not present these results. Please reformat this table according to journal guidelines and include the results by sex on separate lines.
 - b. The results section should exclusively discuss the study results. The manuscript results section as currently written either outline study methods (e.g., lines 122-126) or read as a results discussion section (e.g., lines 133-140). While study results are mixed in there, this section should be revised to provide an unbiased written analysis of the study results. Please move the description of all methods to the Methods section, and citations and comparison of study results with previous research should be included in a manuscript Discussion section. Moreover, no statistics are currently reported in the results section. For example, the written description of Figure 1 results (e.g., line 136) should include if the results were significant or not and at what p-value. The first mention of significance in terms of results appears on line 203 and which references Figure 3, yet results for the prior figures were also significant.
 - c. The results section should be revised to include an in-depth discussion of the findings from the different analytical methods employed. This should include a discussion of the results for the figures and tables, along with the frequencies, statistics, etc. observed.
4. Methods:
 - a. The methods section should include a description of how the tissue samples were collected, stored, and analyzed in the laboratory. While the laboratory was not part of the study, these data should be provided so the reader can fully understand the sample handling and prep methods, and these can impact the results and data. A description of how the patient details included in Table 1 were collected should also be incorporated. Please include details on when tissue samples were collected and when the data included in Table 1 were collected.
 - b. The description of the statistical methods used includes a number of different tests that were done based on the data. Please include clarification regarding what test was used for which results. For example, details on which data were not normally distributed are not included in the written manuscript nor the figures.

Minor Concerns:

1. Only capitalize the first word in the title.
2. Please spell out the first time any acronyms are used in the abstract (e.g., line 39).
3. Please update the cancer statistics in the Introduction and replace all citations corresponding citations with the American Cancer Society’s (ACS) most recently published version of cancer statistics (<https://pmc.ncbi.nlm.nih.gov/articles/PMC11745215/>). In other words, replace the first two references with the 2025 publication.
4. Line 59: The source cited comes from U.S. rather than global statistics or statistics spanning western nations. Please

change 'industrialized world' to the United States.

5. Please ensure there is a space before all citations (e.g., line 61).

6. Place all acronyms in parenthesis and do not bold them (e.g., lines 74-78). Please un-italicize all spelled out acronyms within these lines. Please spell out ω -PUFAs the first time used (line 99).

7. At times, the manuscript uses definitive language where conditional phrasing would be more appropriate. The entire manuscript should be revised to ensure scientific validity and accuracy of the results and statements reported. Example: The use of the word 'strong' in the introduction (line 61), and there should be a comma after cells on line 73.

8. Please review the entire manuscript for grammatical correctness. Example: line 64 should read as lung cancer incidence or rates. Example: cholesterol-rich and nutrition-related should be hyphenated (e.g., line 98). Example: there should be a comma after ω -PUFAs (line 99), and the apostrophe is backwards in patient's (line 100). Example: commas are missing on line 115.

9. Remove the hyphen between the number 3 and PUFAs when referring to omega-3 PUFAs (e.g., line 99).

10. Please do not begin sentences with numbers (e.g., line 611).

11. According to the ACS 2025 cancer statistics, female lung cancer incidence surpassed that of men. Consider including a discussion on these sex differences in the manuscript's discussion section.

Version 1:

Reviewer comments:

Reviewer #1

(Remarks to the Author)

The authors have addressed my comments and improved the manuscript.

Point by point answer to the reviewers

Reviewers' comments:

Reviewer #1 (Remarks to the Author):

The manuscript investigates the interplay between dietary lipids, specifically high cholesterol and omega 3 polyunsaturated fatty acids (PUFAs), and lung cancer progression. Using a combination of human NSCLC tissue analyses and a murine lung cancer model, the authors provide evidence that a high cholesterol diet is associated with increased tumor load, immune evasion (via altered cytokine profiles and regulatory T cell recruitment), and activation of oncogenic pathways. Conversely, omega 3-PUFA supplementation appears to reduce these protumorigenic signals and enhance cytotoxic responses. Overall, the work is relevant to both oncology and nutritional science and may open new avenues for adjunct dietary interventions in NSCLC management.

Major comments

Reviewer:

1. The study relies solely on intravenous injection of LL/2 Luc M38 cells, which may limit the generalizability of the findings. Testing an additional cell line model, such as 393P lung adenocarcinoma cells derived from KrasLA1/+; p53R172HΔG mice, would help validate and strengthen the conclusions.
2. I was surprised that the manuscript did not evaluate the potential impact of omega 3 PUFAs on the efficacy of immune checkpoint inhibitors. Given the prominence of anti PD 1 and CTLA 4 therapies in NSCLC, exploring whether omega 3 PUFAs could synergize with or potentiate these treatments would substantially enhance the study's translational relevance.

Answer:

We appreciate the reviewer's observation regarding the use of a single tumor model. In response, we expanded our experimental approach to include a second model of experimental lung cancer in which L1C2 murine lung adenocarcinoma cells are injected intravenously in BALB/c mice. This experimental model differs both genetically and immunologically from the C57BL/6 LL/2 model. This second model was selected to increase translational relevance and reduce model-specific bias and has been used extensively before by us and others [1, 2]. **Importantly, during this revision, we have performed also a new experiment with this second model of lung cancer enclosing anti-PD-1 (aPD-1) immune checkpoint therapy to evaluate the therapeutic synergy of ω-3 PUFA supplementation in combination with clinically relevant immunotherapy.** Our results confirmed and extended our key findings from the LL/2 model:

- Increased NK cell frequency and activity upon ω-3 PUFA administration, particularly in combination with anti-PD-1 therapy, as indicated by upregulation of the activating receptor NKG2D. (**Figure 6E** in the revised manuscript).
- Reduced accumulation of monocytic MDSCs (Ly6G⁻Ly6C^{high}Gr1⁺CD11b⁺) in the spleen under ω-3 PUFA + anti-PD-1 conditions conditions (**Figure 6F** in the revised manuscript), supporting systemic immune reprogramming.

These results align with and reinforce our original conclusions, demonstrating that dietary ω-3 PUFAs modulate innate immunity and sensitize tumors to immune checkpoint inhibition. Moreover, our

findings are in line with published reports showing that ω -3 PUFAs can reverse immune suppression and enhance NK cell function in various solid tumor contexts.

Accordingly, we have revised the manuscript and figures and marked all newly written or changed paragraphs and figures in blue.

Reviewer:

3. Both the arteriosclerosis diet (AD) and the ω 3 PUFA diet (OD) contain approximately 20% fat; however, the manuscript does not specify the cholesterol content in AD. Without a high fat, cholesterol free control diet, it is difficult to isolate the specific effects of cholesterol from those of fat, saturated fats, or overall caloric density. AD is described as "cholesterol rich" but lacks absolute values (e.g., mg/kg or % w/w cholesterol), and it is unclear whether OD includes any cholesterol. Clarifying these points and including a control group with a high fat, cholesterol free diet would allow for a more precise interpretation of the results.

Answer:

In response to the reviewer's comment we added the total cholesterol value (1520.66mg/kg; 0.15%) of the arteriosclerosis diet in the supplementary data as well as the main manuscript to clarify cholesterol contents in the diets we used. By contrast, the omega-3 rich diet used was equally high in fat but completely free of cholesterol. Additionally, our standard control diet also contained no cholesterol. Thus we are able to interpret the effects of the ω 3 PUFA diet (OD) and the effect of cholesterol in the diet separately.

Reviewer:

4. Although the human sample analysis reveals altered cholesterol metabolism in NSCLC tissues, the absence of detailed dietary intake information, such as food frequency questionnaires or 24 hour dietary recalls, and comprehensive nutritional biomarkers hinders the establishment of a direct link between these molecular changes and patients' dietary habits. Without this information, it remains unclear whether the observed alterations in cholesterol related gene expression and metabolism are driven by dietary lipid intake or by other factors (e.g., underlying disease processes, metabolic conditions, or medication use). Including retrospective or prospective dietary data collection from the human cohort would help correlate nutrient intake with tumor gene expression profiles and clinical outcomes.

Answer:

The detailed information on lipid serum content, which might reflect the diet information on these patients were analyzed extensively and reported in a previous manuscript. [3]

Comprehensive assessment of patients' dietary behavior was not feasible via food frequency questionnaires or 24-hour recalls, given the short duration of hospitalization and the prioritization of surgical and clinical procedures during this time. We also discussed this issue in the manuscript (line 521-522).

Reviewer:

5. While the manuscript demonstrates that dietary omega 3 PUFAs are associated with enhanced CTL cytotoxicity in vitro and reduced induction of regulatory T cells, with concurrent decreases in IL 6Ra and FGF 2 levels in a murine LUAD model, the findings remain largely correlative. The data suggest an association between omega 3 PUFA intake and improved anti tumoral immune responses but do not

fully elucidate the underlying cellular and molecular mechanisms in the tumor immune microenvironment. To strengthen the mechanistic insight, additional experiments are required.

Answer:

Although we agree with the reviewer that our study is only at the beginning and additional studies are needed, we think that this work demonstrate a direct effect of Omega food intake and decrease of inflammatory cytokines (IL-6) and Transcription factors like NFKB. Moreover, we show a direct inhibitory effect of omega 3 on immunosuppressive T regulatory cells. Infact our analysis showed increased populations of **CCR4+** Tregs in lung of mice fed standard diet (SD) or arteriosclerosis diet (AD) when compared to mice fed with Ω -3-rich PUFA diet (OD) or intervention diet. These and additional factors like increased cytotoxic effect against tumor in the supernatants derived from anti-CD3/CD28 challenged lung cells are not correlative but cause effect related to the use of omega 3 diet in mice.

In addition, in human lung adenocarcinoma cell experiments we found an important direct anti metastatic effect of the omega 3 on the human tumor cell line A459 as demonstrated in **Figure 1L/M**

Reviewer:

Comprehensive immunophenotyping of the tumor microenvironment (e.g., using flow cytometry to quantify and characterize tumor infiltrating lymphocyte subsets such as CD8+ T cells, regulatory T cells, NK cells, and myeloid derived suppressor cells) would provide more direct evidence of how dietary omega 3 PUFAs reshape the immune landscape and contribute to anti tumoral activity.

Answer:

We agree with the reviewer's comment regarding the potential involvement of other immunocompetent cell types. In this study, we specifically focused on T cells, given their central role in current immunotherapeutic strategies. Changes observed in T cell populations are reflective of broader alterations within the tumor microenvironment. Furthermore, we have previously demonstrated that cholesterol modulates the immunological synapse between tumor cells and T cells, as detailed in our prior work [3], which included mechanistic analyses based on a cohort of over 100 patients with non-small cell lung cancer (NSCLC).

In response to the reviewers comment, we expanded our flow cytometry-based immunophenotyping efforts in the tumor-bearing mice, with a particular focus on natural killer (NK) cells and monocytic myeloid-derived suppressor cells (M-MDSCs)—two key innate immune populations that are increasingly recognized as important modulators of immunotherapy responsiveness and dietary lipid effects. As previously described, we performed detailed immune profiling in the BALB/c L1C2 model, which we introduced to broaden model diversity and translational relevance (as described in response to Reviewer's Comment #1/#2). This analysis revealed that:

- NK cells (CD3⁻CD335⁺CD49b⁺) were significantly increased in both frequency and activity (as measured by NKG2D expression) in mice fed a ω -3 PUFA-enriched diet, especially when combined with anti-PD-1 therapy.
- M-MDSCs (Ly6G⁻Ly6C^{high}Gr1⁺CD11b⁺) were significantly reduced in the spleens of ω -3 PUFA-treated mice, indicating systemic alleviation of immunosuppressive burden.

While this round of immunophenotyping focused on NK cells and MDSCs—both directly implicated in ω -3 PUFA-mediated immunomodulation in our dataset and the literature—we agree that expanded analysis of T cell subsets, including CD8⁺ T cells and regulatory T cells, would provide additional mechanistic insights. We consider this an important direction for future studies to further

characterize T cell dynamics within the tumor microenvironment in response to dietary lipid modulation. Nonetheless, the newly generated data substantially advance our understanding of how ω -3 PUFAs modulate key innate immune populations and support our conclusion that dietary ω -3 PUFAs enhance anti-tumoral immunity by reshaping the immune landscape.

Minor comments

1. Correct typographical errors in the title (e.g., “faciliated” should be “facilitated”).
2. Ensure consistent use of abbreviations throughout the manuscript, standardizing between “ ω 3 PUFAs” and “omega 3 PUFAs.”

Answer:

We thank the reviewer for their comment. In the revised manuscript, we changed this accordingly.

Amr Khalifa
AIRC-Foundation for Cancer Research Post-Doc Fellow
University of Genoa, Italy

Reviewer #2 (Remarks to the Author):

The study led by Harre et al. aimed to assess the role of dietary lipids, cholesterol, and omega-3 polyunsaturated fatty acids within lung cancer and their potential role in modulating cancer treatment. Below are my comments and suggestions.

Comments for the Author:

Major Concerns:

Reviewer:

1. Study experimental system: This study had two components: an analysis of lung tissue samples from 152 patients and a mice model study, yet there is no indication of these experimental systems in the study title, abstract, or introduction.

Answer:

We apologize about this. In the revised manuscript we inserted these information to the title, abstract and introduction.

Reviewer:

It is not until line 180 that in vivo is mentioned and the word mice is written for the first time on line 179. The manuscript mentions that the study will look at cohorts of patients at the end of the introduction (line 113), yet it is not clear what the multiple cohorts were. The methods section stated that study data were provided by the University of Erlangen's hospital, yet the methods do not mention multiple cohorts. The experimental systems used to study the disease mechanisms should be clearly explained and described in the initial manuscript pages.

Answer:

We revised the manuscript and corrected information regarding the cohort we analyzed and provided a more detailed description of our in vivo study design earlier in the text.

Reviewer:

2. Introduction: The manuscript discusses omega 3-PUFAs but does not provide any details on what these are. As this is a cornerstone of the manuscript, including details on omega PUFAs would be important. Similarly, while the lung tumor microenvironment is included as part of the research objective, it is unclear what a TME actual is. A clear explanation of the concepts central the to study would improve the manuscript overall.

Answer:

As suggested, we added a brief overview of the central concepts to our study, respectively omega PUFAs and the tumor micro environment. (line 113-122, 138-141).

Reviewer:

3. Results:

a. This section should begin with a description of the Table 1 results, for readers to understand the study population and the distribution of its characteristics. While Table 1 is referenced on line 132

and included in the manuscript (line 808), the manuscript does not present these results. Please reformat this table according to journal guidelines and include the results by sex on separate lines.

Answer:

We thank the reviewer for their valuable comment. In response, we have revised the manuscript to include a detailed description of the results presented in Table 1 at the beginning of the respective *Results* section, in order to provide a clearer overview of the study population and the distribution of its clinical characteristics. As per the reviewer's suggestion, we separated Table 1 into Tables 2a and 2b, thereby stratifying the results by gender in distinct tables. The findings are accordingly presented by gender within the Results section.

With regard to the formatting of Table 1, we have carefully reviewed and followed the journal's author guidelines for table preparation to the best of our understanding. Based on the available instructions, we believe the table has been formatted in accordance with the specified requirements. However, we acknowledge that formatting expectations can occasionally vary, and we were unable to find specific guidance in the author instructions that would suggest an alternative format. Should there be additional recommendations or a more detailed template available, we would be grateful for further clarification and happy to revise the table format accordingly to ensure full alignment with the journal's standards.

Reviewer:

b. The results section should exclusively discuss the study results. The manuscript results section as currently written either outline study methods (e.g., lines 122-126) or read as a results discussion section (e.g., lines 133-140). While study results are mixed in there, this section should be revised to provide an unbiased written analysis of the study results. Please move the description of all methods to the Methods section, and citations and comparison of study results with previous research should be included in a manuscript Discussion section. Moreover, no statistics are currently reported in the results section. For example, the written description of Figure 1 results (e.g., line 136) should include if the results were significant or not and at what p-value. The first mention of significance in terms of results appears on line 203 and which references Figure 3, yet results for the prior figures were also significant.

Answer:

We appreciate the reviewer's feedback and the opportunity to clarify our approach. We respectfully believe that the current results section does not contain substantive discussion content, but rather includes concise contextual remarks intended to guide the reader through the rationale for specific analyses. These brief explanatory notes were carefully included to enhance clarity and to illustrate the logical flow of our experimental approach. We feel that this helps the reader in better understanding the motivation behind key comparisons and analytical steps.

We would like to emphasize that the main interpretation of findings and their integration with existing literature are fully reserved for the Discussion section. Nevertheless, we have carefully re-reviewed the manuscript to ensure that methodological details are appropriately placed in the Methods section and that the Results section remains focused on the objective presentation of data. In response to the reviewer's comments, we have also revised the Results section to more clearly indicate when findings reach statistical significance. Additionally, we note that all relevant p-values are reported in the corresponding figure legends to ensure full transparency.

Reviewer:

c. The results section should be revised to include an in-depth discussion of the findings from the different analytical methods employed. This should include a discussion of the results for the figures and tables, along with the frequencies, statistics, etc. observed.

Answer:

In response to the reviewer's request, the Results section of the manuscript has been revised to include a more comprehensive and quantitative discussion of the data from figures and tables. These revisions include references to exact values, sample sizes, and statistical significance for each experimental outcome

Reviewer:

4. Methods:

a. The methods section should include a description of how the tissue samples were collected, stored, and analyzed in the laboratory. While the laboratory was not part of the study, these data should be provided so the reader can fully understand the sample handling and prep methods, and these can impact the results and data. A description of how the patient details included in Table 1 were collected should also be incorporated. Please include details on when tissue samples were collected and when the data included in Table 1 were collected.

Answer:

We thank the reviewer for pointing out the lack of clarity regarding the patient surgery data in Table 1. To improve transparency, we have now included these data in the revised supplementary table 1 and explicitly referenced them in the *Materials and Methods* section. Additionally, we have detailed the tissue storage protocol and clarified the procedures for obtaining clinical patient data, including the use of standardized questionnaires and clinical anamnesis. The process of tissue preparation for qPCR has also been described comprehensively in the *qPCR* subsection.

Reviewer:

b. The description of the statistical methods used includes a number of different tests that were done based on the data. Please include clarification regarding what test was used for which results. For example, details on which data were not normally distributed are not included in the written manuscript nor the figures.

Answer:

In the Methods section, we describe that normality of the data was assessed using the Shapiro–Wilk and D'Agostino–Pearson tests. Based on the distribution and sample size, the following statistical tests were applied: one-way ANOVA with multiple comparisons for normally distributed data; Kruskal–Wallis test for non-normally distributed data or where sample sizes were below $n=5$; unpaired t-tests and Mann–Whitney tests for comparisons between two independent groups; and paired t-tests or Wilcoxon tests for paired data, depending on the distribution. To maintain the clarity and readability of the manuscript, we have provided specific information on the statistical tests used and the distribution of the data in the corresponding figure legends rather than detailing each instance in the main text. We believe this format presents the information in a transparent yet accessible manner, and we hope this approach is acceptable

Reviewer:

Minor Concerns:

1. Only capitalize the first word in the title.
2. Please spell out the first time any acronyms are used in the abstract (e.g., line 39).
3. Please update the cancer statistics in the Introduction and replace all citations corresponding citations with the American Cancer Society's (ACS) most recently published version of cancer statistics (<https://pmc.ncbi.nlm.nih.gov/articles/PMC11745215/>). In other words, replace the first two references with the 2025 publication.
4. Line 59: The source cited comes from U.S. rather than global statistics or statistics spanning western nations. Please change 'industrialized world' to the United States.
5. Please ensure there is a space before all citations (e.g., line 61).
6. Place all acronyms in parenthesis and do not bold them (e.g., lines 74-78). Please un-italicize all spelled out acronyms within these lines. Please spell out ω -PUFAs the first time used (line 99).
7. At times, the manuscript uses definitive language where conditional phrasing would be more appropriate. The entire manuscript should be revised to ensure scientific validity and accuracy of the results and statements reported. Example: The use of the word 'strong' in the introduction (line 61), and there should be a comma after cells on line 73.
8. Please review the entire manuscript for grammatical correctness. Example: line 64 should read as lung cancer incidence or rates. Example: cholesterol-rich and nutrition-related should be hyphenated (e.g., line 98). Example: there should be a comma after ω -PUFAs (line 99), and the apostrophe is backwards in patient's (line 100). Example: commas are missing on line 115.
9. Remove the hyphen between the number 3 and PUFAs when referring to omega-3 PUFAs (e.g., line 99).
10. Please do not begin sentences with numbers (e.g., line 611).
11. According to the ACS 2025 cancer statistics, female lung cancer incidence surpassed that of men. Consider including a discussion on these sex differences in the manuscript's discussion section.

Answer:

We thank the reviewer for their comments that helped us to improve our work. In the revised manuscript, we changed this accordingly.

REFERENCES:

1. Maxeiner, J.H., et al., *A key regulatory role of the transcription factor NFATc2 in bronchial adenocarcinoma via CD8+ T lymphocytes*. *Cancer Res*, 2009. **69**(7): p. 3069-76.
2. Reppert, S., et al., *A role for T-bet-mediated tumour immune surveillance in anti-IL-17A treatment of lung cancer*. *Nat Commun*, 2011. **2**: p. 600.
3. Hartmann, P., et al., *Contribution of serum lipids and cholesterol cellular metabolism in lung cancer development and progression*. *Scientific Reports*, 2023. **13**(1).